# Unveiling Molecular Secrets: An LLM-Augmented Linear Model for Explainable and Calibratable Molecular Property Prediction

## Abstract

Explainable molecular property prediction is essential for various scientific fields, such as drug discovery and material science. Despite delivering intrinsic explainability, linear models struggle with capturing complex, non-linear patterns. Large language models (LLMs), on the other hand, yield accurate predictions through powerful inference capabilities yet fail to provide chemically meaningful explanations for their predictions. This work proposes a novel framework, called *MoleX*, which leverages LLM knowledge to build a simple yet powerful linear model for accurate molecular property prediction with faithful explanations. The core of *MoleX* is to model complicated molecular structure-property relationships using a simple linear model, augmented by LLM knowledge and a crafted calibration strategy. Specifically, to extract the maximum amount of task-relevant knowledge from LLM embeddings, we employ information bottleneck-inspired fine-tuning and sparsity-inducing dimensionality reduction. These informative embeddings are then used to fit a linear model for explainable inference. Moreover, we introduce residual calibration to address prediction errors stemming from linear models' insufficient expressiveness of complex LLM embeddings, thus recovering the LLM's predictive power and boosting overall accuracy. Theoretically, we provide a mathematical foundation to justify *MoleX*'s explainability. Extensive experiments demonstrate that *MoleX* outperforms existing methods in molecular property prediction, establishing a new milestone in predictive performance, explainability, and efficiency. In particular, *MoleX* enables CPU inference and accelerates large-scale dataset processing, achieving comparable performance $300\times$ faster with 100,000 fewer parameters than LLMs. Additionally, the calibration improves model performance by up to 12.7% without compromising explainability. The source code is available at `https://github.com/MoleX2024/MoleX`.

## 1 Introduction

Molecular property prediction, aiming to analyze the relationship between molecular structures and properties, is crucial in various scientific domains, such as computational chemistry and biology (Xia et al., 2024; Yang et al., 2019). Deep learning advancements have significantly improved this field, showcasing the success of AI-driven problem-solving in science. Representative deep models for predicting molecular properties include graph neural networks (GNNs) (Lin et al., 2022; Wu et al., 2023b) and LLMs (Chithrananda et al., 2020; Ahmad et al., 2022). In particular, recently developed LLMs have exhibited remarkable performance by learning chemical semantics from text-based molecular representations, e.g., Simplified Molecular Input Line Entry Systems (SMILES) (Weininger, 1988). By capturing the chemical semantics and long-range dependencies in text-based molecules, LLMs show promising capabilities in providing accurate molecular property predictions (Ahmad et al., 2022). Nevertheless, the black-box nature of LLMs hinders the understanding of their decision-making mechanisms. Inevitably, this opacity prevents people from deriving reliable predictions and insights from these models (Wu et al., 2023a).

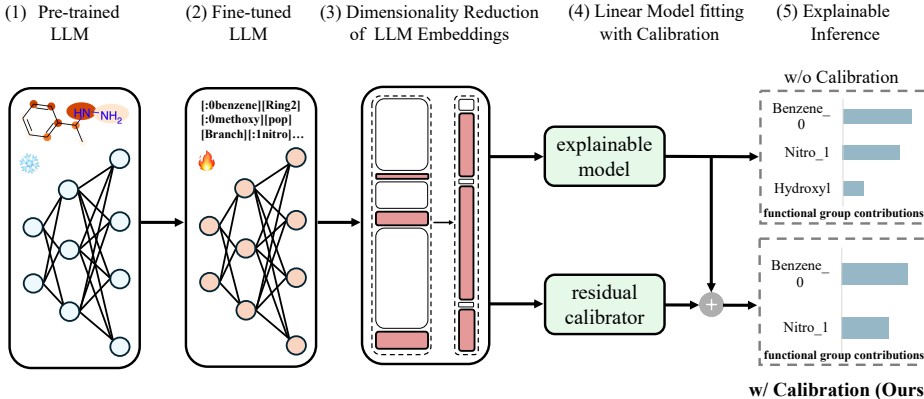

(1) Pre-trained LLM    (2) Fine-tuned LLM    (3) Dimensionality Reduction of LLM Embeddings    (4) Linear Model fitting with Calibration    (5) Explainable Inference

Figure 1: The workflow of *MoleX* includes: (1) using pre-trained ChemBERTa-2, (2) fine-tune it on Group SELFIES (functional group-based molecular representation) with an information bottleneck-inspired objective to produce embeddings with maximum task-relevant information, (3) extract high-dimensional LLM embeddings and apply sparsity-inducing dimensionality reduction to remove redundancy, (4) train a linear model using the preserved task-relevant information, (5) integrate the linear model with a residual calibrator that corrects prediction errors for explainable inference.

To narrow this gap, numerous explainable GNN and LLM methods have been proposed to identify molecular substructures that contribute to specific properties (Xiang et al., 2023; Proietti et al., 2024; Wang et al., 2024). Among these, Lamole (Wang et al., 2024) represents the state-of-the-art LLM-based approach attempting to provide both accurate predictions and chemically meaningful explanations—*chemical concepts-aligned substructures along with their interactions*. However, it still suffers from several flaws: *first*, the attention weights used for explanations do not correlate directly with feature importance (Jain and Wallace, 2019); *second*, it is model-specific due to varying implementations and interpretations of attention mechanisms across models (Voita et al., 2019); and *third*, the provided explanations are local, struggling to approximate global model decisions using established chemical concepts (Liu et al., 2022). Therefore, it is imperative to design a globally explainable method that delivers accurate predictions and identifies contributing substructures with their interactions for molecular property predictions.

We propose a novel framework (illustrated in Figure 1), dubbed *MoleX*, that leverages a linear model augmented with LLM knowledge for explaining *complex, non-linear*

---

**Algorithm 1** Training and Inference of *MoleX*

**Input:** Dataset $\mathcal{S}_D = \{(x^{(i)}, y^{(i)})\}$ where $x^{(i)}$ are input Group SELFIES, $y^{(i)}$ are molecular properties.

1: Split dataset: $\mathcal{S}_D = \mathcal{S}_{\text{train}} \cup \mathcal{S}_{\text{eval}} \cup \mathcal{S}_{\text{test}}$
2: **for** each $x^{(i)}$ in $\mathcal{S}_D$ **do**
3:      Extract $n$-gram feature $x^{(i),\text{ngram}} = \text{N-gram}(x^{(i)})$
4:      Obtain embeddings $e^{(i)} = \text{Extract}(x^{(i),\text{ngram}})$
5:      Reduce dimension $\tilde{x}^{(i)} = \text{EFPCA}(e^{(i)})$
6: **end for**
7: Decompose $\tilde{x}^{(i)}$ into $f_H(\tilde{x}^{(i)})$ and $f_R(\tilde{x}^{(i)})$:
8:      $f_H(\tilde{x}^{(i)})$: explainable features used by $h$
9:      $f_R(\tilde{x}^{(i)})$: residual features used by $r$
10: Train explainable model $h$ by minimizing:

$$h = \arg\min_h \sum_{i \in \mathcal{S}_{\text{train}}} \mathcal{L}\left(h\left(f_H\left(\tilde{x}^{(i)}\right)\right), y^{(i)}\right)$$

11: **for** each $i \in \mathcal{S}_{\text{eval}}$ **do**
12:      Compute residual $y_r^{(i)} = y^{(i)} - h(f_H(\tilde{x}^{(i)}))$
13: **end for**
14: Train residual calibrator $r$ by minimizing:

$$r = \arg\min_r \sum_{i \in \mathcal{S}_{\text{eval}}} \mathcal{L}\left(r\left(f_R\left(\tilde{x}^{(i)}\right)\right), y_r^{(i)}\right)$$

15: **for** each $i \in \mathcal{S}_{\text{test}}$ **do**
16:      Compute the overall prediction:

$$\hat{y}^{(i)} = \text{Aggregate}\left(h(f_H(\tilde{x}^{(i)})), r(f_R(\tilde{x}^{(i)}))\right)$$

17: **end for**

---

molecular structure-property relationships, motivated by its simplicity and global explainability. To capture these complex relationships, *MoleX* extracts informative knowledge from the LLM, which serve as inputs fit to a linear model. Moreover, we design information bottleneck-inspired fine-tuning and sparsity-inducing dimensionality reduction to maximize task-relevant information

in LLM embeddings. Following prior work (Wang et al., 2024), we use Group SELFIES (Cheng et al., 2023)—a text-based representation that partitions molecules into functional groups—as the LLM's input (as shown in appendix A.14). Group SELFIES enables LLMs to tokenize molecules into units of functional groups, aligning with chemical concepts at the substructure level. To quantify functional groups' contributions, we extract n-grams from Group SELFIES and feed them into the LLM, generating embeddings with semantically distinct functional groups for nuanced analysis. Notably, *MoleX*'s simplicity enables global explanations by approximating model behavior across the entire input space, rather than interpreting specific samples.

Although augmented with LLM knowledge, linear models still underfit complex non-linear relationships. To address this, we propose a residual calibration strategy that learns and corrects the linear model's residuals, iteratively bridging the gap between high-dimensional LLM embeddings and linear model's limited expressiveness by calibrating predictions. By iteratively driving residuals toward target values, the residual calibrator calibrates errors and restores the original LLM's predictive power. The linear model, augmented by LLM knowledge and a residual calibrator, achieves excellent predictive performance while retaining the explainability of linear models. In molecular context, the residual calibrator enables *MoleX* to iteratively correct mispredicted functional groups and interactions, aligning predictions with domain expertise and leveraging chemically accurate substructures as explanations. Our contributions are summarized as

1. We propose *MoleX*, which extracts LLM knowledge to build a simple yet powerful linear model that identifies chemically meaningful substructures with their interactions for explainable molecular property predictions.

2. We develop optimization-based methods to maximize and preserve task-relevant information in LLM embeddings and theoretically demonstrate their explainability and validity.

3. We design a residual calibration strategy to correct linear model's prediction errors, improving both predictive and explanation performance.

4. We introduce n-gram coefficients, with a theoretical justification, to assess individual functional group contributions to molecular property predictions.

Experiments across 7 datasets demonstrate that *MoleX* achieves state-of-the-art classification and explanation accuracy while being $300\times$ faster with 100,000 fewer parameters than alternative baselines, highlighting its superiority in predictive performance, explainability, and efficiency.

## 2 RELATED WORK

**Explainable Molecular Property Prediction.** Given that molecules can be naturally represented as graphs, a collection of explainable GNNs have been proposed to explain the relationship between molecular structures and properties (Lin et al., 2021; Pope et al., 2019). However, these atom or bond-level explanations are not chemically meaningful to interpret their sophisticated relationships. Besides, through learning chemical semantics, the transformer-based LLMs can effectively capture interactions among substructures (Wang et al., 2024) and thus demonstrated their potential in understanding text-based molecules (Ross et al., 2022; Chithrananda et al., 2020). However, the opaque decision-making process of LLMs obscures their operating principles, risking unfaithful predictions with severe consequences, especially in high-stakes domains like drug discovery (Chen et al., 2024).

**Explainability Methods for LLMs.** To obtain trustworthy output, various techniques were introduced to unveil the LLM's explainability. The gradient-based explanations analyze the feature importance by computing output partial derivatives with respect to input (Sundararajan et al., 2017). These methods, nevertheless, lack robustness in their explanations due to sensitivity to data perturbations (Kindermans et al., 2019; Adebayo et al., 2018). The attention-based explanations use attention weights to interpret outputs (Hoover et al., 2020). Yet, recent studies challenge their reliability as attention weights may not consistently reflect true feature importance (Jain and Wallace, 2019; Serrano and Smith, 2019). The perturbation-based explanations elucidate model behaviors by observing output changes in response to input alterations (Ribeiro et al., 2016). However, these explanations are unstable due to the randomness of the perturbations (Agarwal et al., 2021). To resolve these issues, we extract informative embeddings from the LLM to fit a linear model for inference. This approach leverages both the LLM's knowledge and the linear model's explainability, offering reliable substructure-level explanations.

## 3 PRELIMINARIES

Let $\mathcal{G} = \{(g^{(i)}, y^{(i)})\}$ be the dataset consisting of molecular graphs $g^{(i)}$ and their corresponding properties $y^{(i)}$. Our goal is to train a model $f$ to map a molecule $g$ to its property $y$, denoted as $f : g \mapsto y$. We first convert each $g^{(i)}$ into Group SELFIES, denoted as $x^{(i)} = \{x_1^{(i)}, \ldots, x_{j^{(i)}}^{(i)}\}$, where $x_j^{(i)}$ is the $j$-th functional group. Specifically, $f$ includes two modules: an explainable model $h$ and a residual calibrator $r$. We decompose $f(x)$, dimensionality reduced LLM embeddings, into $f_H(x)$ and $f_R(x)$, as features used by $h$ and $r$, respectively. Specifically, $f_H(x)$ represents explainable features, capturing variance linked to the property $y$, while residual feature $f_R(x)$ captures the remaining variance. These are projections of $f(x)$ onto orthogonal subspaces, ensuring the contributions of $h$ and $r$ are additive and independent. After $h$ predicts, its residuals are fed into $r$, which boosts performance without incurring any explainability impairment. To learn $h$ and $r$, we freeze the parameters of $h$ and sequentially calibrate its mispredicted samples with the loss $\mathcal{L}$:

$$\min_{h,r} \; \mathbb{E}_{(x,y) \sim \mathcal{D}} \left[ \mathcal{L} \left( h \left( f_H(x) \right) + r \left( f_R(x) \right), y \right) \right], \tag{3.1}$$

where $\mathcal{D}$ is the training dataset. Adapting the approach by Sebastiani (2002), we use n-gram coefficients in the linear model to measure the contributions of decoupled functional groups to molecular properties. Let the functional group $x_j$ takes the coefficient $w_j$ in the linear model; then its contribution score $c_j$ is computed as $c_j = w_j \cdot \text{Embedding}(x_j)$. This allows us to quantify the contribution of the $j$-th functional group to the property $y$ (see our proof of the validity in appendix A.1). For simplicity, we omit the superscript $^{(i)}$ in the following descriptions.

## 4 OUR FRAMEWORK: *MoleX*

As outlined in algorithm 1, *MoleX* operates in two stages: LLM knowledge extraction and LLM-augmented linear model fitting. It extracts n-gram features, generates LLM embeddings, and applies explainable dimensionality reduction. An explainable model $h$ is trained, with a residual calibrator $r$ correcting its prediction errors. During inference, $h$'s predictions are calibrated by $r$, with both models updating their parameters simultaneously to ensure accurate and explainable results.

### 4.1 LLM KNOWLEDGE EXTRACTION WITH IMPROVED INFORMATIVENESS

**Fine-tuning.** To enhance the pre-trained LLM's understanding of functional group-based molecules, we fine-tune it on Group SELFIES data. However, extracting maximally informative LLM embeddings to augment the linear model's expressiveness is still challenging. We address this by integrating the Variational Information Bottleneck (Alemi et al., 2022) into fine-tuning, encouraging the LLM to generate embeddings with maximum task-relevant information, fully leveraging its knowledge. Particularly, given Group SELFIES input $x$, properties $y$, and LLM embeddings $e$, we define $p_0(e)$ as the prior distribution over $e$, and $q_\theta(y \mid e)$ as the variational approximation to the conditional distribution of properties given $e$. The mutual information between $e$ and $y$ is defined as:

$$I(e; y) = \mathbb{E}_{p(e,y)} \left[ \log \frac{p(e,y)}{p(e)p(y)} \right] = \mathbb{E}_{p(e,y)} \left[ \log \frac{p(y \mid e)}{p(y)} \right],$$

and the mutual information between $e$ and $x$ is defined as:

$$I(e; x) = \mathbb{E}_{p(e,x)} \left[ \log \frac{p(e \mid x)}{p(e)} \right] = \mathbb{E}_{p(x)} \left[ D_{\text{KL}} \left( p_\theta(e \mid x) \,\big\|\, p(e) \right) \right].$$

Since the marginal distribution $p(e)$ is intractable, we approximate it with the prior $p_0(e)$. Under this approximation, we use $D_{\text{KL}} \left( p_\theta(e \mid x) \,\big\|\, p_0(e) \right)$ as a tractable surrogate for $I(e; x)$, allowing us to minimize the mutual information between $e$ and $x$. Inspired by Kingma et al. (2015), we approximate encoder $p_\theta(e \mid x)$ by a Gaussian distribution. Let $f_e^\mu(x)$ and $f_e^\Sigma(x)$ be neural networks that output the mean and covariance matrix of latent variable $e$. Then, the encoder is given as:

$$p_\theta(e \mid x) = \mathcal{N} \left( e \,\big|\, f_e^\mu(x), f_e^\Sigma(x) \right).$$

Applying the reparameterization trick, we sample $e$ as:

$$e = f_e^\mu(x) + f_e^\Sigma(x)^{1/2} \cdot \epsilon, \quad \text{where } \epsilon \sim \mathcal{N}(0, I).$$

Putting all these together, we design our training loss as:

$$\mathcal{L}(\theta) = \sum_{(x,y) \in \mathcal{S}_F} \left( \mathbb{E}_{p_\theta(e|x)} \left[ -\log q_\theta(y \mid e) \right] + \beta \cdot D_{\mathrm{KL}} \left( p_\theta(e \mid x) \,\|\, p_0(e) \right) \right), \tag{4.1}$$

where $\beta$ is the tuning parameter between compression and performance, $q_\theta$ is the decoder, and $\mathcal{S}_F$ is the dataset used for fine-tuning. In particular, the first component, $\mathbb{E}_{p_\theta(e|x)} \left[ -\log q_\theta(y \mid e) \right]$, encourages the embeddings $e$ to be informative about $y$ by maximizing their predictive power. The second component, $\beta \cdot D_{\mathrm{KL}} \left( p_\theta(e \mid x) \,\|\, p_0(e) \right)$, regularizes the embeddings to minimize redundant information from $x$, effectively promoting compression.

In essence, this objective ensures the fine-tuned LLM generates embeddings $e$ that capture property-relevant information from $y$ while compressing redundancy in $x$. Grounded in the information bottleneck principle, it produces informative embeddings (see our proof in appendix A.2.)

**Theorem 4.1.** *Let $\mathcal{L}(\theta)$ be the loss defined in eq. (4.1). Under the assumptions of the reparameterization trick and the use of gradient descent, the optimization converges to a local minimum that yields an informative representation $e$ while retaining only relevant information from the task.*

**Embedding Extraction.** To capture individual functional group contributions and contextual information, we extract n-grams from Group SELFIES. To ensure explainability, each n-gram is processed individually by a functional group-level tokenizer, generating fixed-size embeddings. These embeddings are aggregated into a single embedding that encodes the chemical semantics of all n-grams and reflects the knowledge learned by the LLM during training and fine-tuning.

### 4.2 DIMENSIONALITY-REDUCED EMBEDDINGS FOR LINEAR MODEL FITTING

**Dimensionality Reduction.** As the aggregated n-gram embeddings are high-dimensional and noisy, *eliminating the redundancy in them* becomes our new problem. Drawing inspiration from Lin et al. (2016), we design an explainable functional principal component analysis (EFPCA) that leads to effective dimensionality reduction. Accordingly, this preserves a compact yet informative feature set for the linear model. We formulate this dimensionality reduction as an optimization problem with a sparsity-inducing penalty, defined as:

**Definition 4.1** (EFPCA). *Let $X(t)$ be a stochastic process defined on a compact interval $[a, b]$ with mean function $\mu(t) = \mathbb{E}[X(t)]$. Assume that $X(t)$ has a covariance operator $\hat{\mathcal{C}}$ derived from the centered process $X(t) - \mu(t)$. The EFPCA seeks functions $\xi_k(t)$ that maximize the variance explained by the projections of $X(t)$ while promoting sparsity for explainability. Specifically, for each principal component indexed by $k$, the EFPCA solves:*

$$\max_{\xi_k} \left\{ \langle \xi_k, \hat{\mathcal{C}} \xi_k \rangle - \rho_k \, \mathcal{S}(\xi_k) \right\}$$

*subject to $\|\xi_k\|_\gamma^2 = \|\xi_k\|^2 + \gamma \|\mathcal{D}^2 \xi_k\|^2 = 1$ and $\langle \xi_k, \xi_j \rangle_\gamma = 0$ for all $j < k$.*

*Here, $\|\xi_k\|^2 = \int_a^b \xi_k(t)^2 \, dt$ is the squared $L^2$ norm, $\mathcal{D}^2$ denotes the second derivative operator, so $\mathcal{D}^2 \xi_k(t) = \dfrac{d^2 \xi_k(t)}{dt^2}$. The standard $L^2$ inner product is $\langle f, g \rangle = \int_a^b f(t) g(t) \, dt$, and the roughness-penalized inner product is $\langle f, g \rangle_\gamma = \langle f, g \rangle + \gamma \langle \mathcal{D}^2 f, \mathcal{D}^2 g \rangle$, where $\gamma > 0$ balances fit and smoothness. The parameter $\rho_k > 0$ controls sparsity. The function $\mathcal{S}(\xi_k) = \int_a^b \mathbf{1}_{\{\xi_k(t) \neq 0\}} \, dt$ measures the support length of $\xi_k(t)$. The index $k$ specifies the principal components, with $k = 1, 2, \ldots$.*

Since $\xi_k(t)$ is a linear combination of basis functions, we expand it using basis functions $\{\phi_j(t)\}_{j=1}^p$ with local support on sub-intervals $S_j \subset [a, b]$ as $\xi_k(t) = \sum_{j=1}^p a_{kj} \phi_j(t)$, where $a_k = (a_{k1}, \ldots, a_{kp})^\top$ are coefficients to be determined. In this finite-dimensional setting, the support length $\mathcal{S}(\xi_k)$ approximates to $\mathcal{S}(\xi_k) \approx \sum_{j=1}^p \mathbf{1}_{\{a_{kj} \neq 0\}} |S_j|$, which is proportional to the $\ell_0$ "norm"

of $a_k$, $\|a_k\|_0 = \sum_{j=1}^p \mathbf{1}_{\{a_{kj} \neq 0\}}$, assuming equal $|S_j|$. The $\ell_0$ penalty $\rho_k \|a_k\|_0$ thus promotes sparsity by encouraging many coefficients $a_{kj}$ to be zero when $\rho_k$ is large, forcing $\xi_k(t)$ to be zero over extensive portions of $[a, b]$. Zero coefficients mean zero contributions from corresponding basis functions, so the optimization balances maximizing variance while minimizing the number of nonzero coefficients, preserving significant components. As $\phi_j(t)$ have local support, nonzero $a_{kj}$ correspond to specific intervals $S_j$, resulting in $\xi_k(t)$ being nonzero only over certain intervals. Thus, EFPCA produces sparse, explainable principal components due to their localized structure, highlighting regions where the data exhibits significant variation.

In summary, EFPCA offers a framework for explainable principal components, enabling effective dimensionality reduction. By combining a sparsity-inducing penalty with the local support of basis functions, the resulting principal components are sparse and capable of capturing informative features. Therefore, *MoleX* excludes irrelevant functional groups and identifies principal ones from high-dimensional embeddings. We thus formulate the theorem as (see our proof in appendix A.3):

**Theorem 4.2.** *The EFPCA produces sparse FPCs $\xi_k(t)$ that are exactly zero in intervals where the sample curves exhibit minimal variation. Consequently, the FPCs $\xi_k(t)$ are statistically explanatory, facilitating effective dimensionality reduction.*

**Linear Model Fitting.** Applying dimensionality-reduced n-gram embeddings as features, we train a logistic regression model for our classification tasks, which takes the form:

$$h(f_H(x)) = \sigma\left(w^\top f_H(x) + b\right) = \frac{1}{1 + e^{-(w^\top f_H(x) + b)}}, \tag{4.2}$$

where $\sigma$ is the sigmoid function, $w \in \mathbb{R}^n$ is the weight vector, $b \in \mathbb{R}$ is the bias term, and $f_H(x)$ is the explainable features defined in eq. (3.1). The logistic regression is explainable since the log-odds transformation establishes a linear relationship between the features and the target variable, shown as $\log\left(\frac{h(f_H(x))}{1-h(f_H(x))}\right) = w^\top f_H(x) + b$. Differentiating with respect to a feature component $[f_H(x)]_j$ shows that each coefficient $w_j$ quantifies the impact of that feature on the log-odds, shown as $\frac{\partial}{\partial[f_H(x)]_j} \log\left(\frac{h(f_H(x))}{1-h(f_H(x))}\right) = w_j$. Moreover, as $f_H$ is a linear transformation, the chain rule relates changes in the original features to the log-odds, which can be expressed as $\frac{\partial}{\partial x_j} \log\left(\frac{h(f_H(x))}{1-h(f_H(x))}\right) = \sum_{k=1}^n w_k C_{kj}$. Thus, the linearity allows straightforward interpretation of feature impact on the predictions, making logistic regression highly explainable (Hastie et al., 2009).

**Residual Calibration.** The final step of *MoleX* involves training a residual calibrator $r$. With the parameters of the explainable model $h$ frozen, the calibrator corrects mispredicted samples from $h$. By optimizing the objective in eq. (3.1), prediction errors are iteratively fixed, progressively aligning overall predictions with target values. Besides, to maintain explainability, the residual calibrator is designed as a linear model. Specifically, we define the residual calibrator $r$ with weights $w_r \in \mathbb{R}^{d_r}$ corresponding to each residual feature and bias $b_r$:

$$r(f_R(x)) = w_r^\top f_R(x) + b_r.$$

Here, $f_R(x)$ represents the residual features obtained from the decomposition of the feature space $\mathbb{R}^d$ into orthogonal subspaces such that $f(x) = f_H(x) + f_R(x)$ with $f_H(x), f_R(x) \in \mathbb{R}^d$. The vector $f_H(x)$ contains the explainable features used by $h$ and has non-zero components only in the index set $I_H \subseteq \{1, 2, \ldots, d\}$, while $f_R(x)$ contains the residual features used by $r$ and has non-zero components only in the index set $I_R \subseteq \{1, 2, \ldots, d\}$, with $I_H \cap I_R = \emptyset$ and $I_H \cup I_R = \{1, 2, \ldots, d\}$. The orthogonality condition is given by $\langle f_H(x), f_R(x) \rangle = 0$, which holds because the supports of $f_H(x)$ and $f_R(x)$ are disjoint. Then, the overall prediction from $h$ and $r$ is given by:

$$\hat{y}(x) = \underbrace{w_h^\top f_H(x) + b_h}_{\text{Explainable Model Contribution}} + \underbrace{w_r^\top f_R(x) + b_r}_{\text{Residual Calibrator Contribution}},$$

where $w_h, w_r \in \mathbb{R}^d$ are the weight vectors for $h$ and $r$, respectively, with $w_h$ and $w_r$ having non-zero components only in $I_H$ and $I_R$, respectively. The orthogonality and linearity between $f_H(x)$ and $f_R(x)$ guarantee that the contributions from $h$ and $r$ are additive and independent, making the $r$ explainable. Moreover, each feature's impact on the prediction can be directly understood through

the corresponding weights in $w_h$ and $w_r$. Since $f_H(x)$ and $f_R(x)$ are orthogonal, the inner products $w_h^\top f_R(x) = 0$ and $w_r^\top f_H(x) = 0$ vanish. This ensures that $h$ and $r$ do not influence each other's feature contributions, thus preserving the explainability of both models in the combined prediction. Empirically, both $h$ and $r$ update their parameters during prediction error calibration to enhance overall model performance. We formalize the following theorem (see our proof in appendix A.4):

**Theorem 4.3.** *Let $\mathcal{X}$ and $\mathcal{Y}$ be the input and output spaces, respectively. Let $f : \mathcal{X} \to \mathbb{R}^d$ be a pre-trained feature mapping, and let $h : \mathbb{R}^{d_c} \to \mathcal{Y}$ be an explainable linear model operating on the explainable features $f_H(x)$. The residual calibrator $r : \mathbb{R}^{d_r} \to \mathcal{Y}$, defined on the residual features $f_R(x)$, captures the variance not explained by $h$ in an explainable manner, thereby preserving the overall model's explainability.*

**Quantifiable Functional Group Contributions.** As described in section 3, we measure the functional group $x_j$'s contributions to molecular property $y$ using n-gram coefficients. The molecular property $y$ distributes its entire semantic information into individual functional groups $x_j$. Due to the linearity and additivity between $x_j$ and $y$, the scalar coefficient $w_j$ corresponding to $x_j$ in the linear model weighs $x_j$'s contributions to $y$ in terms of chemical semantics. By taking the dot product of $w_j$ and the embedding of $x_j$, we obtain a projection length of the functional group in the direction of weight vector, thus quantifying the impact of that functional group on the molecular property. Quantitatively, the larger the absolute value of an n-gram coefficient, the greater the contribution of the corresponding functional group to property. This metric provides a rigorous interpretation of feature contributions, ensuring unbiasedness and significance through OLS estimation (see our proof in appendix A.1). Using this method, we identify important functional groups from the LLM's complex embedding space. Furthermore, by incorporating n-gram coefficients and identified functional groups into the molecular graph, we can determine whether identified functional groups bond with each other and infer interactions among them. Based on this, *MoleX* reveals chemically meaningful substructures along with their interactions to faithfully explain molecular property predictions.

## 5 EXPERIMENTS

### 5.1 EXPERIMENTAL SETTINGS

**Datasets.** We empirically evaluate *MoleX*'s performance on six mutagenicity datasets and one hepatotoxicity dataset. The mutagenicity datasets include Mutag (Debnath et al., 1991), Mutagen (Morris et al., 2020), PTC family (i.e., PTC-FM, PTC-FR, PTC-MM, and PTC-MR) (Toivonen et al., 2003) and the hepatotoxicity dataset includes Liver (Liu et al., 2015). To demonstrate that *MoleX* can explain molecular properties using chemically meaningful substructures, we introduce the concept of ground truth: substructures verified by domain experts to have significant impacts on molecular properties. The ground truth substructures for six mutagenicity datasets are provided by Lin et al. (2022); Debnath et al. (1991), while those for the hepatotoxicity dataset are provided by Cheng et al. (2023). Further details are available in appendix A.5.

**Evaluation Metrics.** In this study, we evaluate the predictive performance, explainability performance, and computational efficiency of *MoleX*. Particularly, we apply a specific metric to assess each aspect of the model performance. For predictive performance, we define $\frac{1}{I} \sum_{i=1}^{I} \mathbb{I}(y^{(i)} = \hat{y}^{(i)})$ to compute the classification accuracy. For explainability performance, we follow GNNExplainer

(Ying et al., 2019), treating explanations as binary edge classification and using AUC to measure their accuracy. Noteworthily, as LLMs' probabilistic distributions over large vocabularies are incompatible with AUC's binary classification framework, we thus can not offer explanation accuracy for LLMs. For computational efficiency, we evaluate the execution time for each method.

**Baselines.** To extensively compare *MoleX* with different methods, we utilize (1) GNN baselines, including GCN (Kipf and Welling, 2016), DGCNN (Zhang et al., 2018), edGNN (Jaume et al., 2019), GIN (Xu et al., 2018), RW-GNN (Nikolentzos and Vazirgiannis, 2020), DropGNN (Papp et al., 2021), and IEGN (Maron et al., 2018); (2) LLM baselines, including Llama 3.1-8b (Dubey et al., 2024), GPT-4o (Achiam et al., 2023), and ChemBERTa-2 (Ahmad et al., 2022); (3) explainable model baselines, including logistic regression, decision tree (Quinlan, 1986), XGBoost (Chen and Guestrin, 2016), and random forest (Breiman, 2001).

Table 1: Classification accuracy over seven datasets (%). The best results are highlighted in **bold**.

| Methods | Mutag | Mutagen | PTC-FM | PTC-FR | PTC-MM | PTC-MR | Liver |
|---|---|---|---|---|---|---|---|
| GCN (Kipf and Welling, 2016) | 83.4± 0.4 | 77.2± 0.7 | 56.5± 0.3 | 62.7± 0.5 | 58.3± 0.2 | 52.1± 0.6 | 40.6± 0.3 |
| DGCNN (Zhang et al., 2018) | 86.2± 0.2 | 73.7± 0.5 | 56.1± 0.4 | 64.0± 0.8 | 61.8± 0.7 | 57.1± 0.6 | 45.4± 0.9 |
| edGNN (Jaume et al., 2019) | 85.4± 0.6 | 76.5± 0.3 | 58.7± 0.4 | 66.3± 0.7 | 65.2± 0.6 | 55.1± 0.8 | 43.7± 0.4 |
| GIN (Xu et al., 2018) | 86.1± 0.3 | 81.0± 0.5 | 63.4± 0.8 | 67.8± 0.6 | 66.5± 0.4 | 65.5± 0.4 | 45.2± 0.9 |
| RW-GNN (Nikolentzos and Vazirgiannis, 2020) | 88.2± 0.6 | 79.6± 0.2 | 60.5± 0.7 | 63.2± 0.5 | 61.1± 0.4 | 58.2± 0.6 | 42.9± 0.3 |
| DropGNN (Papp et al., 2021) | 90.3± 0.5 | 82.2± 0.3 | 61.4± 0.8 | 65.3± 0.6 | 62.9± 0.2 | 63.5± 0.7 | 46.1± 0.6 |
| IEGN (Maron et al., 2018) | 83.9± 0.4 | 79.3± 0.5 | 61.9± 0.4 | 60.1± 0.3 | 62.1± 0.4 | 60.7± 0.5 | 44.8± 0.8 |
| LLAMA3.1-8b (Dubey et al., 2024) | 67.6± 3.4 | 50.7± 3.6 | 49.6± 2.6 | 46.2± 3.8 | 42.0± 2.8 | 47.5± 2.8 | 42.2± 2.2 |
| GPT-4o (Achiam et al., 2023) | 73.5± 3.6 | 51.2± 0.5 | 52.7± 2.3 | 53.8± 2.9 | 48.8± 2.4 | 53.7± 1.8 | 44.5± 2.5 |
| ChemBERTa-2 (Ahmad et al., 2022) | 87.3± 2.7 | 77.6± 2.2 | 59.2± 1.9 | 64.8± 2.2 | 59.7± 2.8 | 59.8± 2.4 | 46.3± 2.3 |
| Logistic Regression | 58.3± 1.2 | 55.4± 0.8 | 48.4± 1.1 | 48.3± 1.0 | 48.7± 1.1 | 44.9± 1.0 | 32.5± 0.5 |
| Decision Tree (Quinlan, 1986) | 60.8± 1.7 | 58.6± 1.5 | 43.3± 1.0 | 46.1± 0.7 | 47.2± 0.7 | 43.5± 0.5 | 36.9± 0.8 |
| Random Forest (Breiman, 2001) | 64.6± 1.9 | 60.6± 1.5 | 46.9± 1.2 | 51.4± 1.5 | 51.3± 1.8 | 46.4± 1.1 | 34.8± 1.9 |
| XGBoost (Chen and Guestrin, 2016) | 66.9± 1.2 | 67.6± 1.4 | 51.4± 1.3 | 53.1± 1.4 | 55.8± 1.2 | 49.3± 2.1 | 38.5± 1.8 |
| w/o Calibration | 86.1± 2.2 | 74.4± 1.0 | 59.7± 2.1 | 68.9± 1.9 | 69.3± 2.7 | 61.2± 2.4 | 45.0± 2.0 |
| **w/ Calibration (Ours)** | **91.6± 2.0** | **83.7± 0.9** | **64.2± 1.4** | **74.4± 1.9** | **76.4± 1.8** | **68.4± 2.3** | **54.9± 2.4** |

Table 2: Explanation accuracy over seven datasets (%). The best results are highlighted in **bold**.

| Methods | Mutag | Mutagen | PTC-FM | PTC-FR | PTC-MM | PTC-MR | Liver |
|---|---|---|---|---|---|---|---|
| GCN (Kipf and Welling, 2016) | 81.1± 0.2 | 76.4± 0.2 | 65.3± 0.4 | 67.8± 0.7 | 70.8± 0.8 | 65.1± 0.2 | 62.8± 0.2 |
| DGCNN (Zhang et al., 2018) | 86.3± 1.2 | 87.1± 0.5 | 63.0± 1.3 | 57.0± 1.2 | 63.0± 1.3 | 62.3± 0.8 | 67.5± 1.6 |
| edGNN (Jaume et al., 2019) | **94.7± 0.9** | 74.4± 0.7 | 65.9± 0.5 | 64.1± 0.5 | 66.6± 0.7 | 61.4± 0.7 | 63.2± 0.3 |
| GIN (Xu et al., 2018) | 92.1± 0.2 | 75.6± 0.3 | 67.5± 0.6 | 69.2± 0.5 | 68.5± 0.8 | 61.3± 0.5 | 68.3± 0.9 |
| RW-GNN (Nikolentzos and Vazirgiannis, 2020) | 89.9± 0.6 | 76.7± 0.2 | 65.8± 0.3 | 55.5± 0.3 | 66.9± 0.1 | 59.3± 0.2 | 64.7± 0.5 |
| DropGNN (Papp et al., 2021) | 83.4± 0.2 | 77.4± 0.3 | 68.4± 0.2 | 64.7± 0.4 | 63.2± 0.2 | 57.4± 0.7 | 64.5± 0.8 |
| IEGN (Maron et al., 2018) | 82.0± 0.2 | 77.5± 0.2 | 61.6± 0.6 | 62.6± 0.9 | 69.3± 0.7 | 59.1± 0.7 | 66.6± 0.6 |
| Logistic Regression | 59.2± 0.4 | 50.6± 0.9 | 54.4± 0.3 | 47.7± 0.8 | 49.9± 0.7 | 44.3± 0.7 | 53.8± 0.7 |
| Decision Tree (Quinlan, 1986) | 61.2± 0.2 | 55.7± 1.0 | 56.7± 0.8 | 46.4± 1.1 | 48.1± 0.9 | 39.9± 0.8 | 56.4± 1.0 |
| Random Forest (Breiman, 2001) | 66.7± 1.2 | 57.2± 1.2 | 59.9± 1.7 | 55.0± 1.2 | 55.0± 0.8 | 46.6± 1.1 | 60.7± 1.4 |
| XGBoost (Chen and Guestrin, 2016) | 65.2± 1.2 | 61.3± 1.1 | 58.5± 1.8 | 49.4± 1.8 | 51.6± 1.3 | 50.2± 0.8 | 69.0± 1.4 |
| w/o Calibration | 90.0± 0.9 | 77.7± 1.0 | 68.0± 1.7 | 66.6± 1.1 | 62.0± 1.5 | 67.5± 1.5 | 72.0± 2.0 |
| **w/ Calibration (Ours)** | 92.6± 1.7 | **89.0± 1.2** | **77.9± 1.5** | **79.3± 1.4** | **72.3± 1.7** | **73.4± 1.3** | **80.3± 1.4** |

**Implementations.** Our model is pre-trained on the full ZINC dataset (Irwin et al., 2012) using ChemBERTa-2, with $15\%$ of tokens in each input randomly masked. We then fine-tune this model on the Mutag, Mutagen, PTC-FM, PTC-FR, PTC-MM, PTC-MR, and Liver datasets (in Group SELFIES). To evaluate model performance, we compute the average and standard deviation of each metric for each method after 20 rounds of execution. Further details are provided in appendix A.6.

## 5.2 RESULTS

**Predictive Performance.** Table 1 presents a comparison of predictive performance across different methods. *MoleX* outperforms all baselines, showing robustness and generalizability. By combining LLMs with explainable models, it achieves $16.9\%$ and $23.1\%$ higher average accuracy than LLM and explainable model baselines, proving the effectiveness of augmenting explainable models with LLM knowledge. Moreover, by integrating residual calibration, *MoleX* raises the average classification accuracy by $7.0\%$ across seven datasets. Notably, the classification accuracy of our base model, logistic regression, improves by $27.8\%$ after LLM knowledge augmentation and then by an additional $5.5\%$ after residual calibration on the Mutag dataset. Therefore, by maximizing task-relevant semantic information in the LLM knowledge and employing a residual calibration strategy, we enable a simple linear model to achieve predictive performance even superior to that of GNNs and LLMs in molecular property predictions.

**Explainability Performance.** Table 2 summarizes the explanation accuracy of different methods. Be encoding functional group-based molecules, *MoleX* achieves significantly better explainability than baselines. Residual calibration further enhances explainability, improving average accuracy

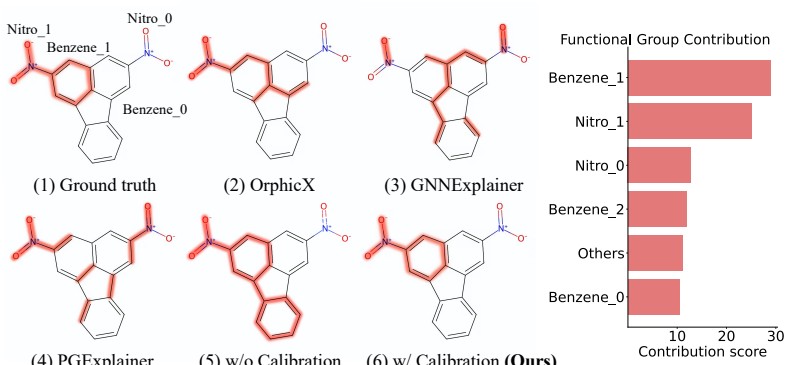

Figure 2: Explanation visualization of a molecule from the Mutag dataset (left), and the contribution scores of the identified functional groups offered by *MoleX* (right).

by 8.8%. It achieves this by iteratively correcting mispredicted functional groups and leveraging chemically accurate ones with their interactions to explain molecular properties. On the Mutag, the explanation accuracy of logistic regression is boosted by 33.4% via LLM knowledge augmentation and residual calibration. Interestingly, while others excel on simpler datasets like Mutag but falter on complex ones, *MoleX* achieves 13.2% higher classification and 16.9% higher explanation accuracy on Liver. It highlights *MoleX*'s capability of representing the complexity of molecular data.

Figure 2 visualizes the explanation for a randomly selected molecule from the Mutag dataset. The ground truth, verified by domain experts, attributes mutagenicity to an aromatic functional group (e.g., benzene ring) bonded with a group like nitro or carbonyl. *MoleX* accurately identifies this substructure, faithfully explaining structure-property relationships. In contrast, other methods identify only individual atoms and bonds, failing to capture chemically meaningful substructures. For example, PGExplainer highlights single atoms from multiple benzene rings, which cannot fully explain molecular properties. Notably, *MoleX* **without** calibration identifies extra elements beyond the ground truth, emphasizing the importance of residual calibration for explanation accuracy. Contribution scores further highlight interactions among functional groups, with the benzene-nitro substructure receiving a high score, demonstrating its role in mutagenicity as an interacting entity. Additional visualizations are provided in appendix A.11.

**Computational Efficiency.** Figure 3 displays the inference time of different methods. Unlike approaches relying on iterative neural network optimization, *MoleX* enables considerably faster inference. It outperforms GNNs (at least $15\times$ faster) and LLMs (at least $120\times$ faster) while achieving higher classification and explanation accuracy. *MoleX* consistently has the lowest inference time across all datasets, highlighting its scalability for real-world applications and large-scale molecular data computations. Furthermore, it reduces GPU memory usage by avoiding iterative parameter updates and storage required in optimization algorithms. This demonstrates how LLM knowledge and residual calibration enhance the linear model's inference power while maintaining explainability and computational efficiency.

## 5.3 ABLATION STUDIES

In this section, we introduce ablation studies on the number of $n$ in n-gram, principal components in EFPCA, training iterations of the residual calibrator, and the selection of the base model.

**Number of $n$ in N-grams.** We empirically evaluate the choice of $n$ for n-grams. As shown in fig. 6, model performance improves as $n$ increases from 1 to 3, then declines for $n$ between 4 and 9. Three out of four datasets show optimal performance at $n = 3$. While larger $n$ captures more contextual semantics, including functional group interactions, excessive $n$ introduces irrelevant information, reducing utility. Further details are in appendix A.10.

**Dimensionality Reduction via EFPCA.** We use EFPCA to reduce the dimensionality of LLM embeddings, producing explainable and compact representations. As shown in fig. 5, cross-validation across four datasets determines the optimal number of principal components, with components beyond 20 contributing minimally to molecular property prediction. Additional components increase

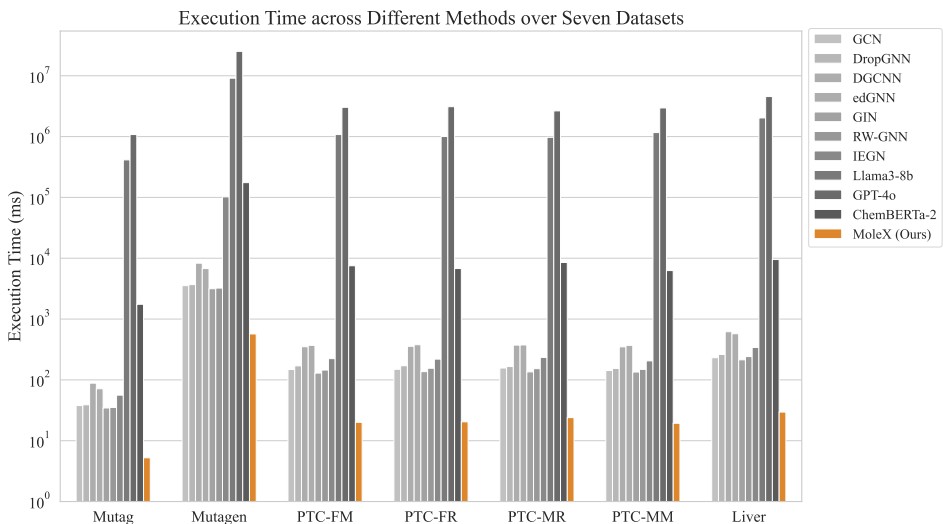

Figure 3: Execution time across different methods over seven datasets. Ours achieves the best inference efficiency.

complexity and reduce explainability. Further details are in appendix A.8. We also evaluate the impact of dimensionality reduction. As shown in table 5, models using only 20 principal components perform within 5% of models using all components, preserving task-relevant information while eliminating redundancy. Additional details are provided in appendix A.9.

**Training Iterations of the Residual Calibrator.** Using the training objective in 3.1, we train a residual calibrator to iteratively correct prediction errors. As shown in fig. 4, model performance improves with more training iterations but declines past a threshold due to overfitting. This highlights the need for an appropriate stopping criterion to balance performance and prevent overfitting. Empirically, the optimal number of iterations is 5. Further details and theoretical justification are provided in appendix A.7.

**Selection of the Base Model.** Other than the logistic regression, we also assess the impact of LLM augmentation using other statistical learning models as base models. Classification and explanation accuracy are presented in table 6 and table 7, respectively. All models augmented with LLM knowledge and residual calibration outperform GNNs and LLMs. More complex models, such as XGBoost and random forest, achieve higher classification and explanation accuracy than simpler models like LASSO. This demonstrates the effectiveness and robustness of LLM augmentation in enhancing model performance. However, increased model complexity often reduces explainability. To balance performance and explainability, we select logistic regression as our base model. Further details are provided in appendix A.12.

## 6 CONCLUSION

This work presents *MoleX*, a framework leveraging LLM knowledge to train a linear model for accurate molecular property predictions with chemically meaningful explanations. Using information bottleneck-inspired fine-tuning and sparsity-based dimensionality reduction, *MoleX* extracts task-relevant knowledge for explainable inference. Furthermore, a residual calibration module further boosts performance by correcting prediction errors. During its inference, *MoleX* precisely reveals crucial substructures with their interactions as explanations. Notably, *MoleX* enjoys the advantage of LLM's predictive power while preserving the linear model's intrinsic explainability. Extensive theoretical and empirical justification demonstrate *MoleX*'s exceptional predictive performance, explainability, and efficiency.

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

# A APPENDIX

## A.1 PROOF OF N-GRAM COEFFICIENTS AS VALID CONTRIBUTION SCORES FOR DECOUPLED N-GRAM FEATURES

In this section, we demonstrate that n-gram coefficients in the linear model can be interpreted as feature contribution scores based on the statistical properties of the linear model.

*Proof.* Suppose $\mathbf{E} \in \mathbb{R}^{n \times d}$ is the matrix of n-gram embeddings, where each row $\mathbf{e}_i^\top$ is the embedding of the $i$-th n-gram. Let $\mathbf{v}_{ij} \in \mathbb{R}^d$ be the embedding of the $j$-th feature in the $i$-th n-gram, and suppose that each n-gram consists of $m$ features (we assume $m$ is a constant across all n-grams for simplicity). Let $c_{ij}$ denote the contribution score of the $j$-th feature in the $i$-th n-gram.

We formulate the following linearity assumptions to ensure the validity of using n-gram coefficients as contribution scores:

- **Linearity.** The relationship between the input embeddings and the output is linear. Namely, for all $i$,
$$y_i = \mathbf{e}_i^\top \mathbf{w}^* + \epsilon_i,$$
where $\mathbf{w}^* \in \mathbb{R}^d$ is the true coefficient vector, and $\epsilon_i$ is the error term.

- **N-gram Embedding Decomposition.** Each n-gram embedding $\mathbf{e}_i$ is the average of its constituent feature embeddings:
$$\mathbf{e}_i = \frac{1}{m} \sum_{j=1}^{m} \mathbf{v}_{ij}.$$

- **Ordinary Least Squares (OLS).** The linear model is estimated using OLS by minimizing the residual sum of squares:
$$\hat{\mathbf{w}} = \arg\min_{\mathbf{w}} \sum_{i=1}^{n} (y_i - \mathbf{e}_i^\top \mathbf{w})^2.$$

- **Error Properties.**

  (a) **Zero Mean Errors.** The errors $\epsilon_i$ have zero mean given the embeddings:
  $$\mathbb{E}[\epsilon_i \mid \mathbf{E}] = 0.$$

  (b) **Homoscedasticity.** The errors have constant variance given the embeddings:
  $$\mathrm{Var}[\epsilon_i \mid \mathbf{E}] = \sigma^2,$$
  where $\sigma^2 > 0$ is a constant.

  (c) **No Autocorrelation.** The errors are uncorrelated with each other:
  $$\mathrm{Cov}[\epsilon_i, \epsilon_j \mid \mathbf{E}] = 0 \quad \text{for } i \neq j.$$

- **Full Rank.** The matrix $\mathbf{E}^\top \mathbf{E}$ is invertible (i.e., $\mathbf{E}$ has full column rank).

We define the contribution score of each decoupled n-gram feature as follows:

**Definition A.1.** *The feature contribution score $c_{ij}$ for the $j$-th feature in the $i$-th n-gram is defined as*
$$c_{ij} = \mathbf{v}_{ij}^\top \hat{\mathbf{w}},$$
*where $\hat{\mathbf{w}}$ is the estimated coefficient vector from the linear model.*

**Lemma A.1** (Prediction as Sum of Feature Contributions)**.** *Under Assumption A.1, the predicted output for the $i$-th n-gram is*
$$\hat{y}_i = \mathbf{e}_i^\top \hat{\mathbf{w}} = \frac{1}{m} \sum_{j=1}^{m} c_{ij}.$$

*Proof.* Using the embedding decomposition and the definition of the contribution scores, we have

$$\hat{y}_i = \mathbf{e}_i^\top \hat{\mathbf{w}}$$

$$= \left( \frac{1}{m} \sum_{j=1}^m \mathbf{v}_{ij} \right)^\top \hat{\mathbf{w}}$$

$$= \frac{1}{m} \sum_{j=1}^m \mathbf{v}_{ij}^\top \hat{\mathbf{w}}$$

$$= \frac{1}{m} \sum_{j=1}^m c_{ij}.$$

This completes the proof. □

**Theorem A.2** (Contribution Scores Quantify Individual Feature Contributions). *Under the Linearity assumption (Assumption A.1), the feature contribution scores $c_{ij}$ quantify the contributions of individual features to the prediction $\hat{y}_i$.*

*Proof.* From Lemma A.1, the predicted value $\hat{y}_i$ is given as the average of the feature contribution scores $c_{ij}$:

$$\hat{y}_i = \frac{1}{m} \sum_{j=1}^m c_{ij}.$$

This equation shows that each feature's contribution score $c_{ij}$ directly influences the prediction $\hat{y}_i$. Therefore, $c_{ij}$ quantifies the contribution of the $j$-th feature in the $i$-th n-gram to the prediction.

This completes the proof. □

Due to the statistical properties of the OLS estimator, we formulate the following theorem:

**Theorem A.3** (Properties of the OLS Estimator). *Under Assumptions A.1–A.1, the OLS estimator $\hat{\mathbf{w}}$ satisfies:*

1. ***Unbiasedness.*** $\mathbb{E}[\hat{\mathbf{w}} \mid \mathbf{E}] = \mathbf{w}^*$.

2. ***Variance-Covariance Matrix.*** $\mathrm{Var}[\hat{\mathbf{w}} \mid \mathbf{E}] = \sigma^2 (\mathbf{E}^\top \mathbf{E})^{-1}$.

3. ***Consistency.*** *As $n \to \infty$, $\hat{\mathbf{w}} \xrightarrow{P} \mathbf{w}^*$.*

*Proof.* We prove each property as follows.

**(1) Unbiasedness:** The OLS estimator is given by

$$\hat{\mathbf{w}} = (\mathbf{E}^\top \mathbf{E})^{-1} \mathbf{E}^\top \mathbf{y}.$$

Substituting $\mathbf{y} = \mathbf{E}\mathbf{w}^* + \boldsymbol{\epsilon}$, we have

$$\hat{\mathbf{w}} = \mathbf{w}^* + (\mathbf{E}^\top \mathbf{E})^{-1} \mathbf{E}^\top \boldsymbol{\epsilon}.$$

Taking expectations conditional on $\mathbf{E}$ and using Assumption A.1(a),

$$\mathbb{E}[\hat{\mathbf{w}} \mid \mathbf{E}] = \mathbf{w}^* + (\mathbf{E}^\top \mathbf{E})^{-1} \mathbf{E}^\top \mathbb{E}[\boldsymbol{\epsilon} \mid \mathbf{E}] = \mathbf{w}^*.$$

**(2) Variance-Covariance Matrix:** The variance conditional on $\mathbf{E}$ is

$$\mathrm{Var}[\hat{\mathbf{w}} \mid \mathbf{E}] = \mathrm{Var}\left( (\mathbf{E}^\top \mathbf{E})^{-1} \mathbf{E}^\top \boldsymbol{\epsilon} \mid \mathbf{E} \right)$$

$$= (\mathbf{E}^\top \mathbf{E})^{-1} \mathbf{E}^\top \mathrm{Var}[\boldsymbol{\epsilon} \mid \mathbf{E}] \mathbf{E} (\mathbf{E}^\top \mathbf{E})^{-1}$$

$$= \sigma^2 (\mathbf{E}^\top \mathbf{E})^{-1},$$

using Assumptions A.1(b) and (c).

**(3) Consistency:** As $n \to \infty$, under the Law of Large Numbers,

$$\frac{1}{n}\mathbf{E}^\top\mathbf{E} \xrightarrow{P} \mathbf{Q},$$

where $\mathbf{Q}$ is positive definite due to Assumption A.1. Additionally,

$$\frac{1}{n}\mathbf{E}^\top\boldsymbol{\epsilon} \xrightarrow{P} \mathbf{0},$$

since $\boldsymbol{\epsilon}$ has zero mean and finite variance. Therefore,

$$\hat{\mathbf{w}} = \mathbf{w}^* + (\mathbf{E}^\top\mathbf{E})^{-1}\mathbf{E}^\top\boldsymbol{\epsilon} \xrightarrow{P} \mathbf{w}^*.$$

This completes the proof. $\qquad\square$

To validate the convergence of the contribution scores, we introduce the asymptotic normality of the OLS estimator.

**Corollary A.1** (Asymptotic Normality). *If the error terms $\boldsymbol{\epsilon}$ are independently and identically normally distributed with mean zero and variance $\sigma^2$, then we have*

$$\sqrt{n}(\hat{\mathbf{w}} - \mathbf{w}^*) \xrightarrow{d} \mathcal{N}\left(\mathbf{0}, \sigma^2\mathbf{Q}^{-1}\right),$$

*where $\mathbf{Q} = \lim_{n\to\infty} \frac{1}{n}\mathbf{E}^\top\mathbf{E}$.*

*Proof.* Under the given conditions, the Central Limit Theorem applies to $\mathbf{E}^\top\boldsymbol{\epsilon}$. Specifically,

$$\sqrt{n}(\hat{\mathbf{w}} - \mathbf{w}^*) = (\mathbf{E}^\top\mathbf{E})^{-1}\mathbf{E}^\top\boldsymbol{\epsilon} = \left(\frac{1}{n}\mathbf{E}^\top\mathbf{E}\right)^{-1}\left(\frac{1}{\sqrt{n}}\mathbf{E}^\top\boldsymbol{\epsilon}\right).$$

As $n \to \infty$, $\frac{1}{n}\mathbf{E}^\top\mathbf{E} \xrightarrow{P} \mathbf{Q}$ and $\frac{1}{\sqrt{n}}\mathbf{E}^\top\boldsymbol{\epsilon} \xrightarrow{d} \mathcal{N}(\mathbf{0}, \sigma^2\mathbf{Q})$. Therefore,

$$\sqrt{n}(\hat{\mathbf{w}} - \mathbf{w}^*) \xrightarrow{d} \mathcal{N}\left(\mathbf{0}, \sigma^2\mathbf{Q}^{-1}\right).$$

This completes the proof. $\qquad\square$

**Lemma A.4** (Variance of $\hat{c}_{ij}$). *The variance of the estimated feature contribution score $\hat{c}_{ij} = \mathbf{v}_{ij}^\top\hat{\mathbf{w}}$ is*

$$\mathrm{Var}[\hat{c}_{ij} \mid \mathbf{E}] = \sigma^2\mathbf{v}_{ij}^\top(\mathbf{E}^\top\mathbf{E})^{-1}\mathbf{v}_{ij}.$$

*Proof.* Since $\hat{c}_{ij}$ is a linear function of $\hat{\mathbf{w}}$, its variance conditional on $\mathbf{E}$ is

$$\begin{aligned}
\mathrm{Var}[\hat{c}_{ij} \mid \mathbf{E}] &= \mathrm{Var}\left(\mathbf{v}_{ij}^\top\hat{\mathbf{w}} \mid \mathbf{E}\right)\\
&= \mathbf{v}_{ij}^\top \mathrm{Var}[\hat{\mathbf{w}} \mid \mathbf{E}]\mathbf{v}_{ij}\\
&= \sigma^2\mathbf{v}_{ij}^\top(\mathbf{E}^\top\mathbf{E})^{-1}\mathbf{v}_{ij},
\end{aligned}$$

using the result from Theorem A.3(2).

This completes the proof. $\qquad\square$

Finally, we demonstrate the statistical significance of the feature contribution scores based on the n-gram coefficients.

**Theorem A.5** (t-Statistic for Feature Contribution Scores). *Under the above assumptions, the t-statistic for testing $H_0 : c_{ij} = 0$ is given by*

$$t_{ij} = \frac{\hat{c}_{ij}}{\mathrm{SE}[\hat{c}_{ij}]} = \frac{\mathbf{v}_{ij}^\top\hat{\mathbf{w}}}{\sigma\sqrt{\mathbf{v}_{ij}^\top(\mathbf{E}^\top\mathbf{E})^{-1}\mathbf{v}_{ij}}}.$$

*Proof.* The standard error of $\hat{c}_{ij}$ is

$$\text{SE}[\hat{c}_{ij}] = \sqrt{\text{Var}[\hat{c}_{ij} \mid \mathbf{E}]} = \sigma \sqrt{\mathbf{v}_{ij}^{\top}(\mathbf{E}^{\top}\mathbf{E})^{-1}\mathbf{v}_{ij}}.$$

Therefore, the t-statistic is

$$t_{ij} = \frac{\hat{c}_{ij}}{\text{SE}[\hat{c}_{ij}]}.$$

Under the null hypothesis $H_0 : c_{ij} = 0$ and assuming normality of the errors, $t_{ij}$ follows a t-distribution with $n - d$ degrees of freedom.

This completes the proof. □

From Theorem A.2, we have shown that the feature contribution scores $c_{ij}$ represent the contributions of individual features to the predictions $\hat{y}_i$. The statistical properties outlined in Theorem A.3 and Lemma A.4 guarantee that these estimates are reliable and that their statistical significance can be assessed.

Therefore, we conclude that each feature's contribution to the prediction can be quantified by its corresponding coefficient in the linear model, enabling us to assess the importance of individual features. By mathematically linking the model coefficients to the feature contributions, we validate the use of these coefficients as measures of feature importance. We also establish that using n-gram coefficients derived from feature embeddings and model coefficients as contribution scores for input features is valid and grounded in the statistical properties of the linear model.

By expressing the predicted output as the sum of individual feature contributions, we effectively decouple the influence of each feature or functional group on the output or molecular property. This decoupling allows us to isolate the effect of each n-gram feature or functional group $x$ on the molecular property $y$. Consequently, the contribution scores $c_{ij}$ provide a quantitative measure of how each functional group impacts the molecular property.

This completes the proof.

□

A.2    PROOF OF THEOREM 4.1 (DEMONSTRATION OF VIB-BASED TRAINING OBJECTIVES)

*Proof.* We demonstrate the Variational Information Bottleneck (VIB) framework, which aims to learn a compressed representation $Z$ of the input variable $X$ that preserves maximal information about the target variable $Y$ while being minimally informative about $X$ itself. This is achieved by optimizing the objective function as follows:

$$\mathcal{L}_{\text{IB}}(\theta) = I(Z; X) - \beta I(Z; Y)$$

where $I(\cdot; \cdot)$ is mutual information, $\beta \geq 0$ is a tuning parameter, and $\theta$ represents the parameters of the encoder. Our goal is to derive a tractable variational lower bound of this objective function that can be optimized using stochastic gradient descent.

**Definition A.2** (Mutual Information). *For random variables $X$ and $Z$ with joint distribution $p(X, Z)$, the mutual information $I(X; Z)$ is defined as*

$$I(X; Z) = \mathbb{E}_{p(X,Z)} \left[ \log \frac{p(X, Z)}{p(X)p(Z)} \right]$$

*Alternatively, it can be expressed as*

$$I(X; Z) = \mathbb{E}_{p(X)} \left[ D_{\text{KL}}(p(Z \mid X) \| p(Z)) \right]$$

**Definition A.3** (Kullback-Leibler Divergence). *For probability distributions $P$ and $Q$ over the same probability space, the KL divergence from $Q$ to $P$ is defined as*

$$D_{\mathrm{KL}}(P\|Q) = \int p(x) \log \frac{p(x)}{q(x)} \, dx = \mathbb{E}_{p(x)} \left[ \log \frac{p(x)}{q(x)} \right]$$

**Definition A.4** (Conditional Entropy). *The conditional entropy $H(Y \mid Z)$ is defined as*

$$H(Y \mid Z) = -\mathbb{E}_{p(Z,Y)} \left[ \log p(Y \mid Z) \right]$$

We then formulate the problem. Let $\mathcal{D} = \{(X_i, Y_i)\}_{i=1}^{N}$ be a dataset of input-output pairs sampled from an unknown distribution $p(X, Y)$. The encoder $p_\theta(Z \mid X)$ parameterizes the conditional distribution of $Z$ given $X$, and the decoder $q_\phi(Y \mid Z)$ parameterizes the conditional distribution of $Y$ given $Z$. Our objective is to optimize the parameters $\theta$ and $\phi$ by maximizing the Information Bottleneck Lagrangian as follows:

$$\mathcal{L}_{\mathrm{IB}}(\theta, \phi) = I(Z; Y) - \beta I(Z; X)$$

However, direct computation of $I(Z; Y)$ and $I(Z; X)$ is intractable. Therefore, we derive variational bounds to make the optimization objective tractable. We start by applying the following lemma:

**Lemma A.6** (Variational Upper Bound on $I(Z; X)$). *The mutual information $I(Z; X)$ can be upper-bounded as*

$$I(Z; X) \leq \mathbb{E}_{p(X)} \left[ D_{\mathrm{KL}}(p_\theta(Z \mid X) \| r(Z)) \right]$$

*where $r(Z)$ is an arbitrary prior distribution over $Z$.*

*Proof.* We start by expressing $I(Z; X)$ as

$$I(Z; X) = \mathbb{E}_{p(X)} \left[ D_{\mathrm{KL}}(p_\theta(Z \mid X) \| p(Z)) \right]$$

Since $p(Z) = \int p_\theta(Z \mid X) p(X) \, dX$ is intractable, we introduce an arbitrary prior $r(Z)$ and consider:

$$I(Z; X) = \mathbb{E}_{p(X)} \left[ D_{\mathrm{KL}}(p_\theta(Z \mid X) \| r(Z)) - D_{\mathrm{KL}}(p(Z) \| r(Z)) \right]$$

Here, we utilize the identity:

$$D_{\mathrm{KL}}(p_\theta(Z \mid X) \| p(Z)) = D_{\mathrm{KL}}(p_\theta(Z \mid X) \| r(Z)) - D_{\mathrm{KL}}(p(Z) \| r(Z))$$

since

$$\mathbb{E}_{p(X)} \left[ D_{\mathrm{KL}}(p_\theta(Z \mid X) \| p(Z)) \right] = \mathbb{E}_{p(X)} \left[ D_{\mathrm{KL}}(p_\theta(Z \mid X) \| r(Z)) \right] - D_{\mathrm{KL}}(p(Z) \| r(Z))$$

Since $D_{\mathrm{KL}}(p(Z) \| r(Z)) \geq 0$, it follows that:

$$I(Z; X) \leq \mathbb{E}_{p(X)} \left[ D_{\mathrm{KL}}(p_\theta(Z \mid X) \| r(Z)) \right]$$

This completes the proof. $\square$

**Lemma A.7** (Variational Lower Bound on $I(Z; Y)$). *The mutual information $I(Z; Y)$ can be lower-bounded as*

$$I(Z; Y) \geq \mathbb{E}_{p(X,Y)} \left[ \mathbb{E}_{p_\theta(Z|X)} \left[ \log q_\phi(Y \mid Z) \right] \right] - H(Y)$$

*Proof.* By the definition of mutual information:

$$I(Z;Y) = H(Y) - H(Y \mid Z) = H(Y) + \mathbb{E}_{p(Z,Y)}\left[\log p(Y \mid Z)\right]$$

Since $p(Y \mid Z)$ is generally intractable, we introduce a variational approximation $q_\phi(Y \mid Z)$ and leverage Jensen's inequality:

$$\mathbb{E}_{p(Z,Y)}\left[\log p(Y \mid Z)\right] \geq \mathbb{E}_{p(Z,Y)}\left[\log q_\phi(Y \mid Z)\right]$$

Therefore:

$$I(Z;Y) \geq H(Y) + \mathbb{E}_{p(Z,Y)}\left[\log q_\phi(Y \mid Z)\right]$$

Rewriting the expectation over $p(Z,Y)$ as an expectation over $p(X,Y)$ and $p_\theta(Z \mid X)$, we have:

$$I(Z;Y) \geq H(Y) + \mathbb{E}_{p(X,Y)}\left[\mathbb{E}_{p_\theta(Z|X)}\left[\log q_\phi(Y \mid Z)\right]\right]$$

Thus:

$$I(Z;Y) \geq \mathbb{E}_{p(X,Y)}\left[\mathbb{E}_{p_\theta(Z|X)}\left[\log q_\phi(Y \mid Z)\right]\right] - H(Y)$$

This completes the proof. $\qquad\square$

Now we can formulate the Variational Information Bottleneck (VIB) objective. By combining Lemmas A.6 and A.7, we obtain a tractable objective function.

**Proposition A.8** (Variational Upper Bound on the Information Bottleneck Objective). *The Information Bottleneck Lagrangian can be upper-bounded by the variational objective function:*

$$\mathcal{L}(\theta, \phi) = \mathbb{E}_{p(X,Y)}\left[\mathbb{E}_{p_\theta(Z|X)}\left[-\log q_\phi(Y \mid Z)\right] + \beta\, D_{\mathrm{KL}}(p_\theta(Z \mid X)\|r(Z))\right]$$

*Proof.* Starting from the original objective:

$$\mathcal{L}_{\mathrm{IB}}(\theta, \phi) = I(Z;X) - \beta I(Z;Y)$$

Applying the upper bound of $I(Z;X)$ from Lemma A.6 and the lower bound of $I(Z;Y)$ from Lemma A.7, we get:

$$\mathcal{L}_{\mathrm{IB}}(\theta, \phi) \leq \mathbb{E}_{p(X)}\left[D_{\mathrm{KL}}(p_\theta(Z \mid X)\|r(Z))\right] - \beta\left(\mathbb{E}_{p(X,Y)}\left[\mathbb{E}_{p_\theta(Z|X)}\left[\log q_\phi(Y \mid Z)\right]\right] - H(Y)\right)$$
$$= \mathbb{E}_{p(X)}\left[D_{\mathrm{KL}}(p_\theta(Z \mid X)\|r(Z))\right] + \beta H(Y) - \beta\,\mathbb{E}_{p(X,Y)}\left[\mathbb{E}_{p_\theta(Z|X)}\left[\log q_\phi(Y \mid Z)\right]\right]$$

Since $H(Y)$ is constant with respect to $\theta$ and $\phi$, we can ignore it for optimization purposes. Thus, we define the variational objective function as:

$$\mathcal{L}(\theta, \phi) = \mathbb{E}_{p(X,Y)}\left[\mathbb{E}_{p_\theta(Z|X)}\left[-\log q_\phi(Y \mid Z)\right] + \beta\, D_{\mathrm{KL}}(p_\theta(Z \mid X)\|r(Z))\right]$$

By minimizing $\mathcal{L}(\theta, \phi)$, we effectively minimize an upper bound on $\mathcal{L}_{\mathrm{IB}}(\theta, \phi)$, satisfying our optimization goal.

This completes the proof. $\qquad\square$

In our fine-tuning stage, since the expectation over $p(X, Y)$ is approximated by empirical samples from the dataset $\mathcal{D}$, and the expectations over $p_\theta(Z \mid X)$ are approximated by Monte Carlo sampling using the reparameterization trick. Thus, the loss function is expressed as (this is a generalized form of our designed loss function shown in (4.1)):

$$\hat{\mathcal{L}}(\theta, \phi) = \frac{1}{N} \sum_{i=1}^{N} \left( -\mathbb{E}_{p_\theta(Z|X_i)} \left[ \log q_\phi(Y_i \mid Z) \right] + \beta\, D_{\mathrm{KL}}(p_\theta(Z \mid X_i) \| r(Z)) \right)$$

To demonstrate convergence, we formulate the following theorem:

**Theorem A.9** (Convergence of Stochastic Gradient Descent). *Under standard assumptions of stochastic optimization (e.g., bounded gradients, appropriate learning rates, smoothness conditions), stochastic gradient descent (SGD) converges to a local minimum of $\hat{\mathcal{L}}(\theta, \phi)$.*

*Proof.* While neural network training is non-convex, empirical and theoretical results in optimization suggest that SGD can converge to critical points (which may be local minima, maxima, or saddle points) provided the loss function is smooth (i.e., continuously differentiable) and the gradients are Lipschitz continuous. Given that $\hat{\mathcal{L}}(\theta, \phi)$ is composed of differentiable functions, and the gradients with respect to $\theta$ and $\phi$ can be computed via backpropagation, convergence to a local minimum is attainable under proper settings of the learning rate and optimization parameters.

This completes the proof. $\square$

We express the following corollary regarding our learned molecular representation after fine-tuning:

**Corollary A.2** (Informative and Compressed Molecular Representation). *At convergence, the learned representation $Z$ satisfies:*

$I(Z; Y)$ *is maximized, and* $I(Z; X)$ *is minimized (subject to the tuning parameter $\beta$)*

*Proof.* By optimizing the variational objective function $\hat{\mathcal{L}}(\theta, \phi)$, we are effectively minimizing an upper bound on $I(Z; X)$ (Lemma A.6) and maximizing a lower bound on $I(Z; Y)$ (Lemma A.7). The trade-off between the two objectives is controlled by $\beta$.

As $\beta$ increases, more emphasis is placed on minimizing $I(Z; X)$, leading to a more compressed representation $Z$ that preserves only the most task-relevant information about $Y$.

This completes the proof. $\square$

Specifically, as the first term in the loss function encourages the embeddings $t$ to be highly predictive of $y$, it intrinsically captures the task-relevant information. Meanwhile, the second term penalizes the complexity of $t$ by forcing it to be close to the prior $p_0(t)$, thereby excluding unnecessary information from $x$. These objectives ensure that the embeddings are both task-relevant and compact, containing minimal spurious data. Additionally, through the derivation of variational bounds and the construction of a tractable objective function, we have shown that minimizing $\mathcal{L}(\theta, \phi)$ allows us to learn a molecular representation $Z$ that captures maximal information about $Y$ while being minimally informative about $X$, in accordance with the Information Bottleneck principle. The optimization of $\mathcal{L}$ via SGD converges to a local minimum under standard optimization assumptions. Therefore, we learn an informative embedding after fine-tuning the pre-trained LLM, and we thus can extract the embedding with improved informativeness.

In conclusion, by framing the fine-tuning within the VIB framework, we derive this approach that balances the essential information for property prediction $y$ with the elimination of irrelevant details from the input molecular representation $x$. This theoretical foundation ensures that our method effectively focuses on extracting the most relevant features needed for accurate predictions.

This completes the proof. $\square$

### A.3 PROOF OF THEOREM 4.2 (EXPLAINABILITY OF EFPCA)

*Proof.* To demonstrate the explainability of the EFPCA method, we will show how the incorporation of a sparsity-inducing penalty and the use of basis functions with local support lead to functional principal components (FPCs) that are both sparse and localized, enhancing interpretability.

First, we formulate the EFPCA as an optimization problem. The EFPCA seeks to find FPCs $\xi_k(t)$ that maximize the variance of the projections of the centered stochastic process $X(t) - \mu(t)$ onto $\xi_k(t)$, while promoting sparsity for explainability. Specifically, for each principal component indexed by $k$, we solve:

$$\max_{\xi_k} \left\{ \langle \xi_k, \hat{\mathcal{C}} \xi_k \rangle - \rho_k \, \mathcal{S}(\xi_k) \right\} \tag{A.1}$$

subject to the normalization constraint:

$$\|\xi_k\|_\gamma^2 = \|\xi_k\|^2 + \gamma \left\| \mathcal{D}^2 \xi_k \right\|^2 = 1, \tag{A.2}$$

and the orthogonality constraints:

$$\langle \xi_k, \xi_j \rangle_\gamma = 0 \quad \text{for all } j < k. \tag{A.3}$$

Here $\hat{\mathcal{C}}$ is the empirical covariance operator of the centered process $X(t) - \mu(t)$, defined by $\hat{\mathcal{C}} f = \int_a^b \hat{c}(t, s) f(s) \, ds$, where $\hat{c}(t, s)$ is the empirical covariance function. $\langle f, g \rangle = \int_a^b f(t) \, g(t) \, dt$ is the standard $L^2$ inner product. $\|f\|^2 = \langle f, f \rangle$ is the squared $L^2$ norm. $\mathcal{D}^2 f = \dfrac{d^2 f(t)}{dt^2}$ denotes the second derivative of $f(t)$. $\|\mathcal{D}^2 f\|^2 = \langle \mathcal{D}^2 f, \mathcal{D}^2 f \rangle$ penalizes the roughness of $f(t)$. $\gamma > 0$ is a smoothing parameter balancing variance explanation and smoothness. $\langle f, g \rangle_\gamma = \langle f, g \rangle + \gamma \langle \mathcal{D}^2 f, \mathcal{D}^2 g \rangle$ is the roughness-penalized inner product. $\mathcal{S}(\xi_k) = \int_a^b \mathbf{1}_{\{\xi_k(t) \neq 0\}} \, dt$ measures the length of the support of $\xi_k(t)$, promoting sparsity. $\rho_k > 0$ controls the sparsity of $\xi_k(t)$. $k$ is the index of the principal component, with $k = 1, 2, \ldots$.

Then, we construct an expansion of $\xi_k(t)$ using basis functions with local support. Let $\{\phi_j(t)\}_{j=1}^p$ be a set of basis functions that have local support on the interval $[a, b]$, such as B-spline basis functions. Each $\phi_j(t)$ is nonzero only over a subinterval $S_j \subset [a, b]$. We express $\xi_k(t)$ as a linear combination of these basis functions:

$$\xi_k(t) = \sum_{j=1}^p a_{kj} \phi_j(t), \tag{A.4}$$

where $a_k = (a_{k1}, a_{k2}, \ldots, a_{kp})^\top$ is the coefficient vector for the $k$-th principal component. We substitute the expansion (A.4) into the optimization problem (A.1). To express the objective function and constraints in terms of $a_k$, we compute the variance explained by $\xi_k(t)$:

$$\langle \xi_k, \hat{\mathcal{C}} \xi_k \rangle = \left\langle \sum_{i=1}^p a_{ki} \phi_i, \hat{\mathcal{C}} \sum_{j=1}^p a_{kj} \phi_j \right\rangle = \sum_{i=1}^p \sum_{j=1}^p a_{ki} a_{kj} \langle \phi_i, \hat{\mathcal{C}} \phi_j \rangle.$$

We define the matrix $\mathbf{Q} \in \mathbb{R}^{p \times p}$ with entries $Q_{ij} = \langle \phi_i, \hat{\mathcal{C}} \phi_j \rangle$, so the variance term becomes $a_k^\top \mathbf{Q} a_k$. The sparsity-inducing term $\mathcal{S}(\xi_k)$ approximates to:

$$\mathcal{S}(\xi_k) \approx \sum_{j=1}^p \mathbf{1}_{\{a_{kj} \neq 0\}} |S_j|,$$

assuming negligible overlap between the supports of different $\phi_j(t)$, where $|S_j|$ is the length of the support of $\phi_j(t)$. If the supports are of equal length or normalized, we can consider $\mathcal{S}(\xi_k) \propto \|a_k\|_0$, where $\|a_k\|_0 = \sum_{j=1}^{p} \mathbf{1}_{\{a_{kj} \neq 0\}}$ counts the number of nonzero coefficients.

Therefore, the objective function becomes:

$$\text{Objective:} \quad a_k^\top \mathbf{Q} a_k - \rho_k \|a_k\|_0. \tag{A.5}$$

We have the roughness-penalized norm is:

$$\|\xi_k\|_\gamma^2 = \langle \xi_k, \xi_k \rangle + \gamma \langle \mathcal{D}^2 \xi_k, \mathcal{D}^2 \xi_k \rangle = a_k^\top \mathbf{G} a_k,$$

where $\mathbf{G} = \mathbf{G}_0 + \gamma \mathbf{G}_2$, with $\mathbf{G}_0$ having entries $(\mathbf{G}_0)_{ij} = \langle \phi_i, \phi_j \rangle$, and $\mathbf{G}_2$ having entries $(\mathbf{G}_2)_{ij} = \langle \mathcal{D}^2 \phi_i, \mathcal{D}^2 \phi_j \rangle$. Thus, the normalization constraint becomes:

$$a_k^\top \mathbf{G} a_k = 1. \tag{A.6}$$

Additionally, the orthogonality constraints with respect to the roughness-penalized inner product are given as:

$$\langle \xi_k, \xi_j \rangle_\gamma = a_k^\top \mathbf{G} a_j = 0, \quad \text{for all } j < k.$$

Combining these, the optimization problem becomes:

$$\max_{a_k} \left\{ a_k^\top \mathbf{Q} a_k - \rho_k \|a_k\|_0 \right\} \tag{A.7}$$

subject to:

$$a_k^\top \mathbf{G} a_k = 1, \quad \text{and} \quad a_k^\top \mathbf{G} a_j = 0 \quad \text{for all } j < k. \tag{A.8}$$

The term $\rho_k \|a_k\|_0$ in the objective function is an $\ell_0$ penalty that promotes sparsity in the coefficient vector $a_k$. When $\rho_k$ is large, the optimization favors solutions with fewer nonzero coefficients, effectively selecting only the most significant basis functions. We define the index set of nonzero coefficients:

$$\mathcal{I}_k = \{j \mid a_{kj} \neq 0\}. \tag{A.9}$$

The principal component $\xi_k(t)$ then simplifies to:

$$\xi_k(t) = \sum_{j \in \mathcal{I}_k} a_{kj} \phi_j(t). \tag{A.10}$$

Since each $\phi_j(t)$ has support only on $S_j$, the support of $\xi_k(t)$ is given by:

$$\text{supp}(\xi_k) = \bigcup_{j \in \mathcal{I}_k} S_j. \tag{A.11}$$

Thus, $\xi_k(t)$ is exactly zero outside these intervals, and nonzero only over regions where significant variation is captured by the selected basis functions. The localization of $\xi_k(t)$ enhances explainability in several ways:

- **Identification of Significant Intervals.** The nonzero coefficients $a_{kj}$ correspond to basis functions whose supports $S_j$ cover intervals where the data exhibits important features. This directly highlights regions of interest in the functional data.

- **Simplification of Interpretation.** By reducing the number of nonzero coefficients, $\xi_k(t)$ becomes simpler and easier to interpret, focusing on key patterns in the data.

- **Exclusion of Irrelevant Information.** The sparsity induced by the $\ell_0$ penalty effectively filters out noise and redundant information, ensuring that only meaningful variations are considered.

Moreover, the roughness penalty $\gamma\|\mathcal{D}^2\xi_k\|^2$ ensures that $\xi_k(t)$ remains smooth within its support, avoiding overfitting and maintaining the functional integrity of the principal components. The parameter $\gamma$ balances the trade-off between fitting the data closely and keeping the principal components smooth.

In the context of high-dimensional embeddings from LLMs, the EFPCA method effectively reduces dimensionality while enhancing explainability. By promoting sparsity, it preserves only the most informative features associated with the task, filtering out task-irrelevant information present in the embeddings. The localized structure of $\xi_k(t)$ allows for direct interpretation of the components in terms of specific intervals or features in the data.

In conclusion, the incorporation of a sparsity-inducing $\ell_0$ penalty and the use of basis functions with local support in the EFPCA framework lead to principal components that are both sparse and localized. This results in FPCs $\xi_k(t)$ that are nonzero only over intervals where the data contains significant variation, making them intrinsically explainable. The optimization framework balances variance maximization, sparsity, and smoothness, yielding components that facilitate effective dimensionality reduction while providing clear insights into the underlying functional data. In our implementation, we maintain statistically significant features in an explainable manner, ensuring that the dimensionality reduction aids in both performance and interpretability.

This completes the proof. $\qquad\square$

## A.4    PROOF OF THEOREM A.10 (EXPLAINABILITY OF RESIDUAL CALIBRATION)

*Proof.* We demonstrate that the residual calibrator $r$ is explainable when combined with the explainable linear model $h$, under the conditions of linearity and orthogonality.

Let $\mathcal{X}$ and $\mathcal{Y}$ be the input and output spaces, respectively. Let $f : \mathcal{X} \to \mathbb{R}^d$ be a pre-trained feature mapping that extracts features from the inputs $x \in \mathcal{X}$. We decompose the feature vector $f(x)$ into two components:

$$f(x) = f_H(x) + f_R(x),$$

where $f_H(x), f_R(x) \in \mathbb{R}^d$ are the explainable and residual features, respectively. The vector $f_H(x)$ contains the explainable features used by the explainable model $h$, and has non-zero components only in the index set $I_H \subseteq \{1, 2, \ldots, d\}$. Similarly, $f_R(x)$ contains the residual features used by the residual calibrator $r$, and has non-zero components only in the index set $I_R \subseteq \{1, 2, \ldots, d\}$, with $I_H \cap I_R = \emptyset$ and $I_H \cup I_R = \{1, 2, \ldots, d\}$. To ensure orthogonality between $f_H(x)$ and $f_R(x)$, we observe that their supports are disjoint, implying that their inner product is zero:

$$\langle f_H(x), f_R(x)\rangle = \sum_{i=1}^{d}[f_H(x)]_i \cdot [f_R(x)]_i = 0,$$

since for each $i$, at least one of $[f_H(x)]_i$ or $[f_R(x)]_i$ is zero. The explainable model $h : \mathbb{R}^d \to \mathcal{Y}$ is defined as a linear model operating on $f_H(x)$:

$$h(f_H(x)) = w_h^\top f_H(x) + b_h,$$

where $w_h \in \mathbb{R}^d$ is the weight vector with non-zero components only in $I_H$, and $b_h \in \mathbb{R}$ is the bias term. Similarly, the residual calibrator $r : \mathbb{R}^d \to \mathcal{Y}$ is defined as a linear model operating on $f_R(x)$:

$$r(f_R(x)) = w_r^\top f_R(x) + b_r,$$

where $w_r \in \mathbb{R}^d$ is the weight vector with non-zero components only in $I_R$, and $b_r \in \mathbb{R}$ is the bias term. The overall prediction from $h$ and $r$ is given by:

$$\hat{y}(x) = h(f_H(x)) + r(f_R(x)) = w_h^\top f_H(x) + b_h + w_r^\top f_R(x) + b_r.$$

We define the combined weight vector $w = w_h + w_r \in \mathbb{R}^d$ and combined bias $b = b_h + b_r$, so the prediction simplifies to:

$$\hat{y}(x) = w^\top f(x) + b.$$

Due to the orthogonality of $f_H(x)$ and $f_R(x)$, and the disjoint supports of $w_h$ and $w_r$, the cross terms vanish:

$$w_h^\top f_R(x) = \sum_{i \in I_H} [w_h]_i [f_R(x)]_i = 0, \quad w_r^\top f_H(x) = \sum_{i \in I_R} [w_r]_i [f_H(x)]_i = 0,$$

since $[w_h]_i = 0$ for $i \notin I_H$ and $[f_R(x)]_i = 0$ for $i \in I_H$, and similarly for $w_r$ and $f_H(x)$. This ensures that $h$ and $r$ do not influence each other's feature contributions, thus preserving the explainability of both models in the combined prediction. To illustrate how $r$ captures the variance not explained by $h$ in an explainable manner, consider that the residual calibrator $r$ corrects mispredicted samples from $h$ by fitting to the residuals $y - h(f_H(x))$. By optimizing the objective:

$$\min_r \mathbb{E}_{(x,y) \sim \mathcal{D}} \left[ \mathcal{L} \left( h(f_H(x)) + r(f_R(x)), \, y \right) \right],$$

where $\mathcal{D}$ is the data distribution and $\mathcal{L}$ is a suitable loss function (e.g., mean squared error), the residual calibrator $r$ learns to model the remaining variance in $y$ that $h$ does not capture. The linearity of $r$ ensures that its contribution to the prediction is transparent and explainable. Each residual feature $[f_R(x)]_i$ contributes to $\hat{y}(x)$ proportionally to its corresponding weight $[w_r]_i$:

$$\frac{\partial \hat{y}(x)}{\partial [f_R(x)]_i} = [w_r]_i.$$

Similarly, for the explainable features, we have:

$$\frac{\partial \hat{y}(x)}{\partial [f_H(x)]_i} = [w_h]_i.$$

This allows us to directly understand each feature's impact on the prediction. Furthermore, during training, both $h$ and $r$ can update their parameters to enhance overall model performance. The orthogonality condition allows us to optimize $w_h$ and $w_r$ separately. Considering a convex and differentiable loss function $\ell(\hat{y}, y)$, the gradients with respect to $w_h$ and $w_r$ are:

$$\nabla_{w_h} \mathcal{L} = \mathbb{E}_{(x,y)} \left[ \ell' \left( \hat{y}(x), y \right) f_H(x) \right], \quad \nabla_{w_r} \mathcal{L} = \mathbb{E}_{(x,y)} \left[ \ell' \left( \hat{y}(x), y \right) f_R(x) \right],$$

where $\ell'$ denotes the derivative of $\ell$ with respect to its first argument. Since $f_H(x)$ and $f_R(x)$ have disjoint supports, the inner product $f_H(x)^\top f_R(x) = 0$, and thus the updates to $w_h$ and $w_r$ do not interfere with each other. We formalize these observations in the following theorem:

**Theorem A.10.** *Let $\mathcal{X}$ and $\mathcal{Y}$ be the input and output spaces, respectively. Let $f : \mathcal{X} \rightarrow \mathbb{R}^d$ be a pre-trained feature mapping, and let $h : \mathbb{R}^d \rightarrow \mathcal{Y}$ be an explainable linear model operating on the explainable features $f_H(x)$. The residual calibrator $r : \mathbb{R}^d \rightarrow \mathcal{Y}$, defined on the residual features $f_R(x)$, captures the variance not explained by $h$ in an explainable manner, thereby preserving the overall model's explainability.*

*Proof of Theorem A.10.* As established, the combined model's prediction is:

$$\hat{y}(x) = h(f_H(x)) + r(f_R(x)) = w_h^\top f_H(x) + b_h + w_r^\top f_R(x) + b_r.$$

The orthogonality of $f_H(x)$ and $f_R(x)$, along with the disjoint supports of $w_h$ and $w_r$, ensures that the cross terms vanish, shown as $w_h^\top f_R(x) = 0, \quad w_r^\top f_H(x) = 0$. Therefore, the combined prediction simplifies to sum of individual contributions from $h$ and $r$. To understand how $r$ captures the unexplained variance, consider the total variance of $y$ decomposed into the variance explained by $h$ and the residual variance:

$$\text{Var}(y) = \text{Var}\left(h(f_H(x))\right) + \text{Var}\left(y - h(f_H(x))\right) + 2\,\text{Cov}\left(h(f_H(x)),\, y - h(f_H(x))\right).$$

However, since $y - h(f_H(x))$ is uncorrelated with $h(f_H(x))$ under certain conditions, the covariance term becomes zero, leading to:

$$\text{Var}(y) = \text{Var}\left(h(f_H(x))\right) + \text{Var}\left(y - h(f_H(x))\right).$$

The residual calibrator $r$ models the residual $y - h(f_H(x))$, aiming to minimize $\text{Var}\left(y - h(f_H(x)) - r(f_R(x))\right)$. Since $r$ is linear and operates on $f_R(x)$, and given that $f_H(x)$ and $f_R(x)$ are orthogonal, the variance captured by $r(f_R(x))$ does not overlap with that captured by $h(f_H(x))$. This additive property ensures that the total variance explained by the combined model is:

$$\text{Var}\left(h(f_H(x)) + r(f_R(x))\right) = \text{Var}\left(h(f_H(x))\right) + \text{Var}\left(r(f_R(x))\right),$$

due to the independence arising from orthogonality. The explainability of $r$ is preserved because:

- **Transparency:** The linearity of $r$ allows us to interpret the contribution of each residual feature directly through its weight in $w_r$.

- **Non-Interference:** Orthogonality guarantees that $r$ does not affect the interpretability of $h$, as they operate on separate feature subsets.

- **Predictive Enhancement:** $r$ enhances the predictive performance by capturing additional patterns in the data that $h$ alone cannot explain.

Moreover, from a functional analysis perspective, the projection operators $P_H$ and $P_R$ associated with $f_H(x)$ and $f_R(x)$ satisfy $P_H + P_R = I_d$, where $I_d$ is the identity matrix in $\mathbb{R}^d$. This confirms that the entire feature space is covered by the combined subspaces, and there is no loss of information in the decomposition. Furthermore, considering the operator norms of $h$ and $r$:

$$\|h\|_{\text{op}} = \sup_{\|f_H(x)\|=1} |h(f_H(x))|, \quad \|r\|_{\text{op}} = \sup_{\|f_R(x)\|=1} |r(f_R(x))|,$$

we can analyze the stability and boundedness of both models. The boundedness of $h$ and $r$ ensures that small changes in the input features lead to proportionally small changes in the predictions, which is desirable for model robustness and interpretability. Thus, $r$ captures the variance not explained by $h$ in an explainable manner, preserving the overall model's explainability. This completes the proof of Theorem A.10. The final step of *MoleX* involves training the residual calibrator $r$. With the parameters of the explainable model $h$ frozen (or updated separately due to orthogonality), the calibrator corrects mispredicted samples from $h$. By optimizing the objective:

$$\min_r \mathbb{E}_{(x,y)\sim\mathcal{D}}\left[\mathcal{L}\left(h(f_H(x)) + r(f_R(x)), y\right)\right],$$

prediction errors are iteratively fixed, progressively aligning overall predictions with target values. The design of $r$ as a linear model and its orthogonality with $h$ ensure that explainability is maintained while enhancing model performance. Moreover, each feature's impact on the prediction can be directly understood through the corresponding weights in $w_h$ and $w_r$. Since $f_H(x)$ and $f_R(x)$ are orthogonal, and their weight vectors $w_h$ and $w_r$ have disjoint supports, we have:

$$\frac{\partial \hat{y}(x)}{\partial [f(x)]_i} = \begin{cases} [w_h]_i, & \text{if } i \in I_H, \\ [w_r]_i, & \text{if } i \in I_R. \end{cases}$$

This explicit form provides clear interpretability of the model's predictions, allowing practitioners to understand and trust the contributions of individual features. Thus, under the conditions of linearity and orthogonality, the residual calibrator $r$ preserves explainability when combined with $h$. The combined model benefits from improved predictive accuracy while retaining transparency, satisfying both performance and interpretability objectives.

This completes the proof. $\square$

## A.5 DATASET DETAILS

We use six mutagenicity datasets and one hepatotoxicity dataset. The mutagenicity datasets are: Mutag (Debnath et al., 1991), Mutagen (Morris et al., 2020), PTC-FM (Toivonen et al., 2003), PTC-FR (Toivonen et al., 2003), PTC-MM (Toivonen et al., 2003), PTC-MR (Toivonen et al., 2003), and the hepatotoxicity dataset is the Liver (Liu et al., 2015). Followed by Morris et al. (2020), we list the summary statistics of these datasets as

Table 3: Summary statistics of seven datasets

| Dataset | Mutag | Mutagen | PTC-FM | PTC-FR | PTC-MM | PTC-MR | Liver |
|---|---|---|---|---|---|---|---|
| Samples | 188 | 4337 | 349 | 351 | 336 | 344 | 587 |
| Classes | 2 | 2 | 2 | 2 | 2 | 2 | 3 |
| Ground truth | 120 | 724 | 58 | 49 | 51 | 61 | 187 |

Note: Ground truth refers to the number of annotated samples in each dataset.

The ground truth indicates the true molecular substructures that impact molecular properties. As verified by Lin et al. (2022); Debnath et al. (1991), the ground truth substructures for six mutagenicity datasets consist of an aromatic group, such as a benzene ring, bonded with another functional group, such as methoxy, oxhydryl, nitro, or carboxyl groups (note that ground truth exists only for the mutagenic class). For the Liver dataset, the ground truth annotated by chemists are: fused tricyclic saturated hydrocarbon moiety, hydrazines, arylacetic acid, sulfonamide moiety, aniline moiety, a class of proton pump inhibitor drugs, acyclic bivalent sulfur moiety, acyclic di-aryl ketone moiety, para oxygen and nitrogen di-substituted benzene ring, a relatively small number of com- pounds in the expanded LiverTox dataset, halogen atom bonded to a $sp^3$ carbon, and fused tricyclic structural moiety. A detailed illustration of Liver's ground truth are provided by Liu et al. (2015).

## A.6 IMPLEMENTATION DETAILS

Our model is pre-trained on all data in the ZINC dataset (over 230 million compounds) using ChemBERTa-2, with $15\%$ (default setting) of tokens in each input randomly masked. We extract

all functional groups in the ZINC dataset as the vocabulary to expand the LLM's tokenizer so that the fine-tuned LLM can better encode functional group-level inputs. We then fine-tune this model on Mutag, Mutagen, PTC-FM, PTC-FR, PTC-MM, PTC-MR, and Liver datasets. The fine-tuning is conducted on $1\times$ NVIDIA RTX3090 GPU for about 3 hours. The detailed hyperparameters with their values are given in table 4. For experiments on model performance, we employ chain-of-thought prompting for the molecular property prediction tasks on LLMs.

| Hyperparameter | Value |
|---|---|
| learning rate | 1e-5 |
| batch size | 128 |
| epochs | 30 |
| weight decay | 0.01 |
| gradient clipping | 1.0 |
| warmup proportion | 0.06 |
| max sequence length | 1024 |
| optimizer | AdamW |
| dropout rate | 0.1 |
| gradient accumulation steps | 1 |
| mixed precision training | True |

Table 4: Hyperparameters and their values we used for fine-tuning

We offer the pseudo code to explain our fine-tuning procedure as shown in algorithm 2.

### A.7 DOES THE RESIDUAL CALIBRATOR IMPROVES MODEL PERFORMANCE BY TRAINING WITH MORE ITERATIONS?

We employ the training objective in 3.1 to learn a residual calibrator that iteratively corrects samples the linear model fails to predict accurately. We empirically study how training iterations influence the overall model predictions. As shown in fig. 4, we visualize the model performance on the Mutag, Mutagen, PTC-MR, and Liver datasets under different numbers of training iterations. As training iterations increase, model performance improves significantly until reaching a threshold. This suggests that more iterations on our designed loss lead to better performance. After the threshold, the model overfits the data, resulting in performance degradation. Therefore, increasing the number of training iterations helps improve model performance. Empirically, we found that 5 iterations yield optimal performance. A theoretical demonstration shows that training with multiple iterations increases model performance until a threshold, after which it declines, as follows.

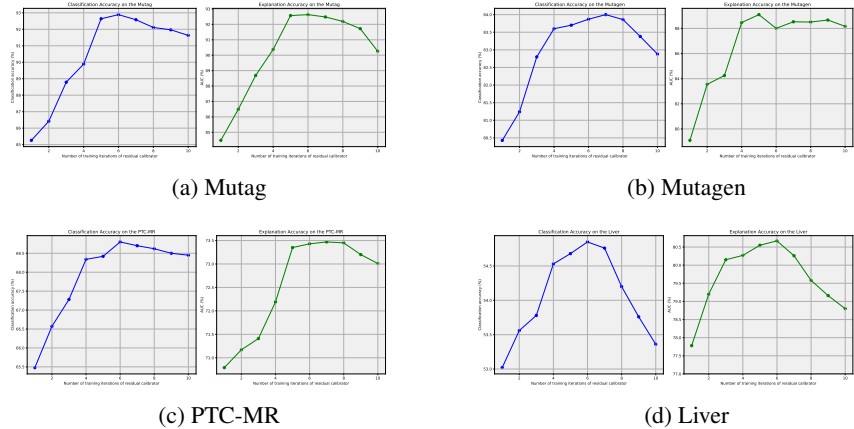

(a) Mutag             (b) Mutagen

(c) PTC-MR             (d) Liver

Figure 4: The model performance with different training iterations of the residual calibrator

**Problem Setup.** Given the objective the residual calibrator minimized during training:

---

**Algorithm 2** Fine-tuning LLM with Group SELFIES

---

**Input:** Fine-tuning dataset $\mathcal{S}_F = \{(x_i, y_i)\}$ where $x_i$ are Group SELFIES, $y_i$ are molecular properties.
**Input:** Initialize ChemBERTa-2 model parameters $\theta$.
**Input:** Prior distribution $p_0(t) = \mathcal{N}(0, I)$.
**Input:** Learning rate $\eta$ and trade-off parameter $\beta$.

1: **while** not converged **do**
2:     **for** each mini-batch $\mathcal{B} \subset \mathcal{S}_F$ **do**
3:         **for** each $(x_i, y_i) \in \mathcal{B}$ **do**
4:             Compute encoder mean and covariance:

$$\mu_i = f_e^\mu(x_i), \quad \Sigma_i = f_e^\Sigma(x_i)$$

5:             Sample $\epsilon_i \sim \mathcal{N}(0, I)$
6:             Generate embedding using reparameterization trick:

$$t_i = \mu_i + \Sigma_i^{1/2} \cdot \epsilon_i$$

7:             Compute decoder loss:

$$\mathcal{L}_{\text{dec}}(i) = -\log q_\theta(y_i | t_i)$$

8:             Compute KL divergence:

$$\mathcal{L}_{\text{KL}}(i) = D_{\text{KL}} \left( p_\theta(t_i | x_i) \,\|\, p_0(t) \right)$$

9:             Compute total loss:

$$\mathcal{L}_i = \mathcal{L}_{\text{dec}}(i) + \beta \cdot \mathcal{L}_{\text{KL}}(i)$$

10:         **end for**
11:         Compute batch loss:

$$\mathcal{L}_\mathcal{B} = \frac{1}{|\mathcal{B}|} \sum_{i \in \mathcal{B}} \mathcal{L}_i$$

12:         Update model parameters:

$$\theta \leftarrow \theta - \eta \cdot \nabla_\theta \mathcal{L}_\mathcal{B}$$

13:     **end for**
14: **end while**

---

$$\min_{h,r} \; \mathbb{E}_{(x,y)\sim\mathcal{S}_{\text{train}}} \left[ \mathcal{L}\left(h(f_H(x)) + r(f_R(x)), \; y\right) \right], \tag{A.12}$$

where $\mathcal{S}_{\text{train}}$ is the empirical distribution of the training data and $\mathcal{L} : \mathbb{R} \times \mathbb{R} \to \mathbb{R}_{\geq 0}$ is a convex, differentiable loss function, e.g., the squared loss $\mathcal{L}(\hat{y}, y) = \frac{1}{2}(\hat{y} - y)^2$. We demonstrate that: initially, as the residual calibrator $r$ is trained, the model's performance on unseen data improves, i.e., the generalization loss decreases. Beyond a certain threshold, further minimization of the training loss leads to overfitting, where the generalization loss starts to increase, and prediction accuracy on unseen data degrades.

*Proof.* We aim to demonstrate that learning the residual calibrator $r$ with multiple training iterations initially improves the model accuracy, but after a certain training threshold, continued minimization of the training loss leads to overfitting, leading to the predictive accuracy on unseen data decline.

Let $\mathcal{X}$ and $\mathcal{Y}$ be the input and output spaces, respectively. Consider a feature extraction function $f : \mathcal{X} \to \mathbb{R}^d$ that maps inputs to a $d$-dimensional feature space. We assume that $f$ can be decomposed into two components:

$$f(x) = f_H(x) + f_R(x),$$

where $f_H(x) \in \mathbb{R}^{d_c}$ represents the explainable features used by the explainable model $h$, and $f_R(x) \in \mathbb{R}^{d_r}$ represents the residual features used by the residual calibrator $r$, with $d = d_c + d_r$. We assume that the feature components $f_H(x)$ and $f_R(x)$ are orthogonal, which means:

$$\langle f_H(x), f_R(x) \rangle = 0 \quad \text{for all } x \in \mathcal{X}.$$

The explainable model $h : \mathbb{R}^{d_c} \to \mathbb{R}$ is defined as a linear model:

$$h(f_H(x)) = W_h^\top f_H(x) + b_h,$$

where $W_h \in \mathbb{R}^{d_c}$ and $b_h \in \mathbb{R}$ are the weights and bias of $h$. The residual calibrator $r : \mathbb{R}^{d_r} \to \mathbb{R}$ is also defined as a linear model:

$$r(f_R(x)) = W_r^\top f_R(x) + b_r,$$

where $W_r \in \mathbb{R}^{d_r}$ and $b_r \in \mathbb{R}$ are the weights and bias of $r$. Due to the orthogonality of $f_H(x)$ and $f_R(x)$, the overall prediction model becomes:

$$\hat{y}(x) = h(f_H(x)) + r(f_R(x)) = W_h^\top f_H(x) + W_r^\top f_R(x) + b_h + b_r.$$

Our objective is to minimize the expected loss:

$$\mathcal{L}(W_h, W_r, b_h, b_r) = \mathbb{E}_{(x,y)\sim\mathcal{D}} \left[ \ell\left(\hat{y}(x), y\right) \right],$$

where $\ell\left(\hat{y}(x), y\right)$ is a convex and differentiable loss function, such as the squared loss $\ell(\hat{y}, y) = \frac{1}{2}(\hat{y} - y)^2$, and $\mathcal{D}$ is the data distribution. We begin by considering the training loss over a finite training dataset $\{(x_i, y_i)\}_{i=1}^n$:

$$\mathcal{L}_{\text{train}}(W_h, W_r, b_h, b_r) = \frac{1}{n} \sum_{i=1}^n \ell\left(\hat{y}(x_i), y_i\right).$$

Initially, when $r$ is untrained or minimally trained, the model may be underfitting, and both the training loss $\mathcal{L}_{\text{train}}$ and generalization loss $\mathcal{L}_{\text{gen}}$ are high. By updating $W_r$ and $b_r$ via gradient descent to minimize $\mathcal{L}_{\text{train}}$, we have the updates:

$$W_r^{(t+1)} = W_r^{(t)} - \eta \nabla_{W_r} \mathcal{L}_{\text{train}}(W_h, W_r^{(t)}, b_h, b_r^{(t)}),$$

$$b_r^{(t+1)} = b_r^{(t)} - \eta \nabla_{b_r} \mathcal{L}_{\text{train}}(W_h, W_r^{(t)}, b_h, b_r^{(t)}),$$

where $\eta > 0$ is the learning rate, and $t$ denotes the iteration number. Since $\ell$ is convex and differentiable, these updates ensure that the training loss decreases:

$$\mathcal{L}_{\text{train}}^{(t+1)} \leq \mathcal{L}_{\text{train}}^{(t)}.$$

During this phase, $r$ captures genuine patterns in the residual features $f_R(x)$ that are not explained by $h$. Consequently, the generalization loss decreases as well:

$$\mathcal{L}_{\text{gen}}^{(t+1)} \leq \mathcal{L}_{\text{gen}}^{(t)},$$

where $\mathcal{L}_{\text{gen}}(W_h, W_r, b_h, b_r) = \mathbb{E}_{(x,y) \sim \mathcal{D}} \left[ \ell \left( \hat{y}(x), y \right) \right].$

However, as training continues, $W_r$ and $b_r$ may begin to fit the noise or idiosyncrasies specific to the training data, especially if the model has a high capacity (i.e., $d_r$ is large relative to $n$). The fitting capacity of $r$ allows it to minimize $\mathcal{L}_{\text{train}}$ further, but this comes at the cost of increasing complexity.

To formalize this, we consider the concept of Rademacher complexity $\mathfrak{R}_n(\mathcal{H})$ for the hypothesis class $\mathcal{H}$ associated with $r$. The Rademacher complexity provides a measure of the model's ability to fit random noise in the data. The generalization error can be bounded as:

$$\mathcal{L}_{\text{gen}}(W_h, W_r, b_h, b_r) \leq \mathcal{L}_{\text{train}}(W_h, W_r, b_h, b_r) + 2\mathfrak{R}_n(\mathcal{H}) + \delta,$$

where $\delta$ is a constant dependent on the loss function and confidence level. As $\|W_r\|$ increases due to continued training, $\mathfrak{R}_n(\mathcal{H})$ increases, reflecting the higher complexity of $r$. This leads to circumstances that:

$$\mathcal{L}_{\text{train}}^{(t+1)} < \mathcal{L}_{\text{train}}^{(t)} \quad \text{but} \quad \mathcal{L}_{\text{gen}}^{(t+1)} > \mathcal{L}_{\text{gen}}^{(t)} \quad \text{for } t \geq t^*,$$

where $t^*$ is the iteration threshold beyond which overfitting occurs.

For linear models, the Rademacher complexity can be bounded by:

$$\mathfrak{R}_n(\mathcal{H}) \leq \frac{B\|W_r\|}{\sqrt{n}},$$

where $B = \sup_{x \in \mathcal{X}} \|f_R(x)\|$. As $\|W_r\|$ increases, $\mathfrak{R}_n(\mathcal{H})$ increases, leading to a wider generalization gap. This increase in model complexity without a corresponding increase in true predictive power causes the model to generalize poorly on unseen data, despite the training loss decreasing. This phenomenon is a bias-variance trade-off: the variance increases significantly due to overfitting, outweighing any small reductions in bias achieved by further minimizing the training loss.

In conclusion, while initial training of the residual calibrator $r$ improves model accuracy by reducing both the training loss and the generalization loss, continued training beyond a certain threshold leads to overfitting. The residual calibrator begins to model noise in the training data, increasing its complexity and causing the generalization loss to increase. This results in a decline in prediction accuracy on unseen data, suggesting the importance of strategies such as early stopping or regularization to prevent overfitting.

This completes the proof. □

## A.8 HOW TO CHOOSE THE OPTIMAL NUMBER OF PRINCIPAL COMPONENTS?

To empirically determine the optimal number of principal components for our implementation, we compare model performance metrics (classification accuracy and explanation accuracy) across four datasets under different numbers of principal components. As shown in fig. 5, both metrics tend to converge as the number of principal components exceeds 20. This indicates that when the number of components surpasses 20, the contribution of additional components to molecular property prediction becomes trivial. In this scenario, adding more components produces diminishing marginal benefits while significantly increasing model complexity, which in turn reduces explainability. Therefore, we choose the top 20 principal components to explain the variance in molecular properties, seeking for a balance between performance and explainability.

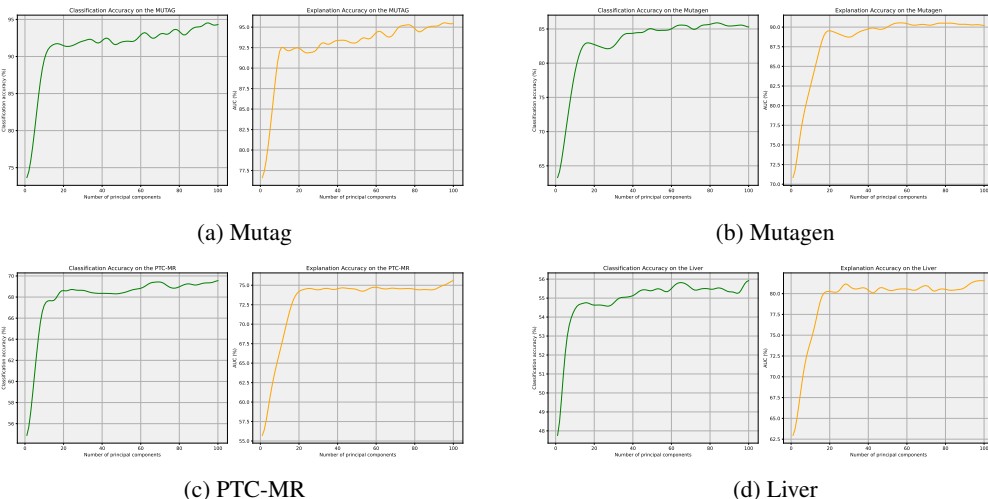

(a) Mutag          (b) Mutagen

(c) PTC-MR          (d) Liver

Figure 5: Optimal number of principal components

## A.9 DOES EFPCA EFFECTIVELY WORKS?

In addition to the analysis in appendix A.8, we demonstrate that the dimensionality reduction by EFPCA effectively preserves the most explanatory components. We compare the model performance across seven datasets *with and without dimensionality reduction*. As shown in table 5, when using only 20 PCs, the model performance improves by no more than $5\%$ compared to using all 384 components (i.e., no dimensionality reduction). This indicates that EFPCA effectively preserves the most task-relevant and important information in LLM embeddings while excluding noisy components. These preserved components achieve comparable performance to the models with all components while being significantly simpler and more explainable. This showcases the success of our dimensionality reduction in maintaining model performance while enhancing explainability.

| Dataset | Classification Accuracy ($\%$) | Explanation Accuracy ($\%$) |
|---------|-------------------------------|-----------------------------|
| Mutag   | $94.9_{\pm1.6}$ | $96.1_{\pm3.0}$ |
| Mutagen | $86.4_{\pm1.4}$ | $91.2_{\pm1.6}$ |
| PTC-FR  | $78.7_{\pm1.2}$ | $82.7_{\pm1.7}$ |
| PTC-FM  | $68.1_{\pm1.5}$ | $81.1_{\pm2.0}$ |
| PTC-MR  | $70.5_{\pm1.7}$ | $76.5_{\pm2.6}$ |
| PTC-MM  | $80.9_{\pm2.7}$ | $75.3_{\pm2.2}$ |
| Liver   | $57.3_{\pm1.6}$ | $83.8_{\pm1.9}$ |

Table 5: Model performance *without* EFPCA over seven datasets

## A.10 DOES THE CHOICE OF $n$ IN N-GRAM MAKES A DIFFERENCE?

We compare the different values of $n$ in n-gram via cross-validation based on our two evaluation metrics, classification accuracy and explanation accuracy. The results in fig. 6 suggest an overall trend that as $n$ goes from 1 to 3, both classification accuracy and explanation accuracy improve; as $n$ goes from 4 to 9, both classification accuracy and explanation accuracy drop. On the four datasets we used for experiments, three of them show that good model performance can be achieved when $n$ is taken to be 3. As $n$ grows from small to large, it encourages the model to capture more contextual semantics, including interactions between functional groups, which allows for a significant improvement in prediction. When $n$ exceeds a certain threshold, irrelevant or even toxic information emerges from the captured contextual information (i.e., irrelevant long-range dependencies), making the overall model utility gradually decreases.

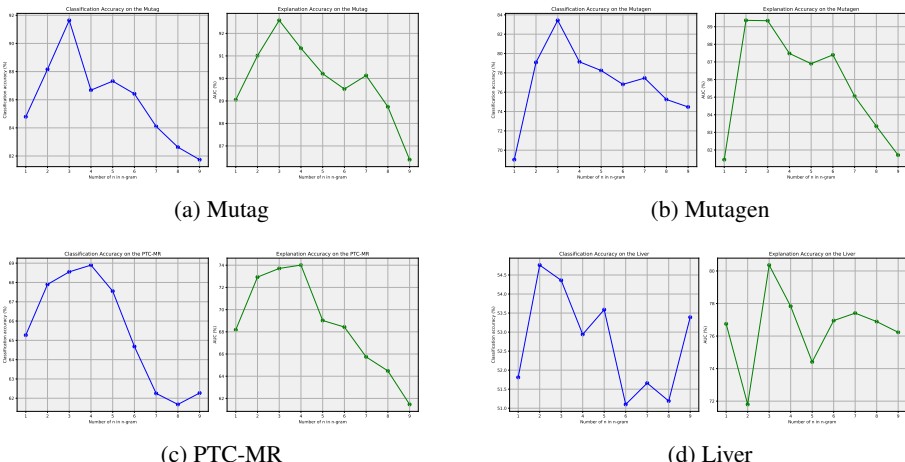

(a) Mutag  (b) Mutagen

(c) PTC-MR  (d) Liver

Figure 6: The choice of $n$ in n-gram on the Mutag, Mutagen, PTC-MR, and Liver datasets

## A.11 MORE EXPLANATION VISUALIZATIONS

We randomly select one sample from each of the six remaining datasets and provide explanation visualizations based on *MoleX*. Specifically, fig. 7, fig. 8, fig. 9, fig. 10, fig. 11, and fig. 12 display the samples selected from the Mutagen, PTC-FM, PTC-MM, PTC-FR, PTC-MR, and Liver datasets, respectively. On the left, we compare molecular substructures identified by different methods, with ground truth showing expert-validated substructures influencing molecular properties. Red marks on the molecular graph highlight key components identified by each method. We compare with three baselines: OrphicX (Lin et al., 2022), GNNExplainer (Ying et al., 2019), and PGExplainer (Luo et al., 2020), as well as *MoleX* with and without residual calibration (w/ denotes *with* and w/o denotes *without*). On the right, we show *MoleX*'s n-gram contribution scores (0–100) for functional groups, with higher scores indicating greater influence on molecular properties.

Taking fig. 7 as an example, *MoleX* precisely identifies the ground truth substructures for the sample from the Mutagen dataset. Specifically, *MoleX* highlights the benzene ring bonded with an amino group on the upper left as vital substructures to explain the molecule's mutagenicity. The contribution scores computed by *MoleX* indicate that the benzene ring has the highest contribution to molecular properties, followed by the amino group. This aligns with the ground truth that a benzene ring bonded with an amino group leads to mutagenicity (Lin et al., 2022; Debnath et al., 1991). Therefore, *MoleX* accurately captures the important functional groups (i.e., the benzene ring and the amino group) and the interaction between them, revealing their precise bonding. As the ground truth indicates, only the bonded benzene and amino group together impact the molecular properties. In contrast, other methods provide only atom or bond-level explanations and fail to discover important functional groups as a whole. They identify only a few atoms and bonds in the benzene or amino group and fail to capture the interaction between these two functional groups. Consequently, these atom or bond-level explanations are insufficiently faithful in explaining molecular properties,

as individual atoms or bonds have limited impact on overall molecular properties (Mirghaffari et al., 2021). The explanation visualizations for samples from other datasets also demonstrate *MoleX*'s effectiveness in identifying important substructures and their interactions, aligning with chemical concepts to explain molecular property predictions.

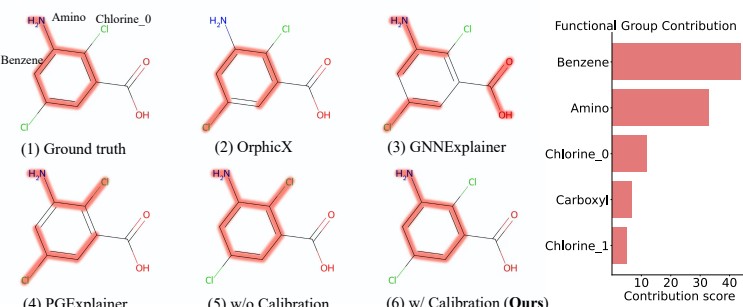

Figure 7: Explanation visualization of a molecule from the Mutagen dataset (left), and contribution scores of the identified functional groups offered by *MoleX* (right).

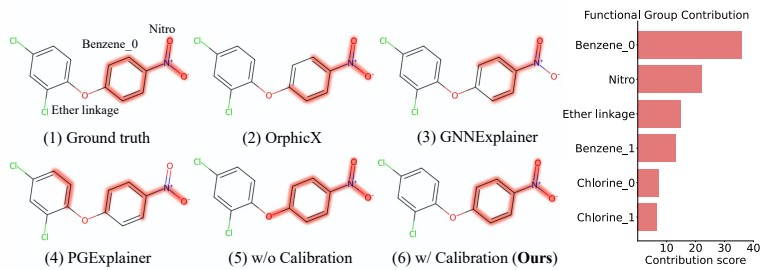

Figure 8: Explanation visualization of a molecule from the PTC-FM dataset (left), and contribution scores of the identified functional groups offered by *MoleX* (right).

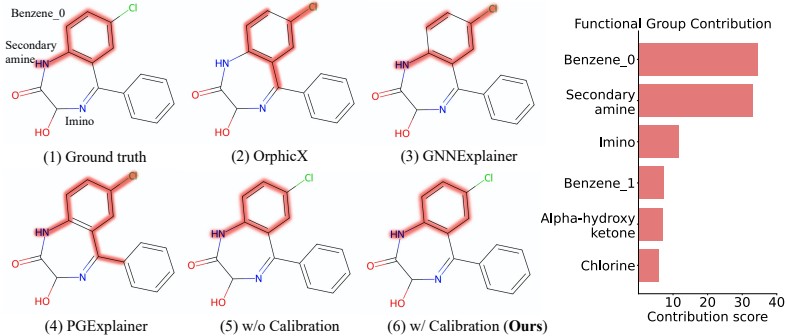

Figure 9: Explanation visualization of a molecule from the PTC-MM dataset (left), and contribution scores of the identified functional groups offered by *MoleX* (right).

## A.12 CAN OTHER STATISTICAL LEARNING MODELS BE AUGMENTED WITH THE LLM KNOWLEDGE?

In addition to the linear model, we augment various statistical learning models with the LLM knowledge and test them on seven datasets. The classification accuracy and explanation accuracy are shown in table 6 and table 7, respectively. Other linear models, such as ridge regression, LASSO, and linear discriminant analysis, achieve comparable performance to *MoleX* and showcase the generalizability of LLM knowledge augmentation on linear models. Additionally, the polynomial re-

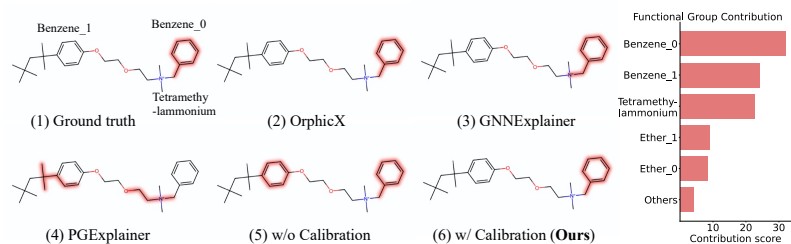

Figure 10: Explanation visualization of a molecule from the PTC-FR dataset (left), and contribution scores of the identified functional groups offered by *MoleX* (right).

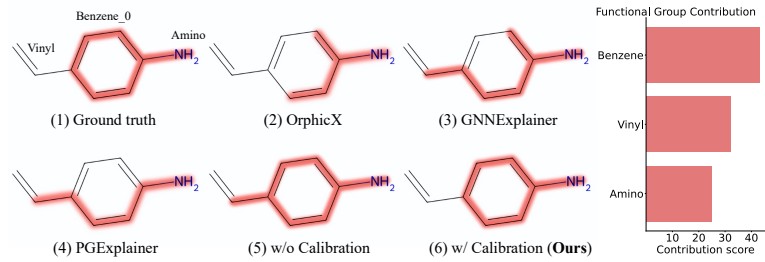

Figure 11: Explanation visualization of a molecule from the PTC-MR dataset (left), and contribution scores of the identified functional groups offered by *MoleX* (right).

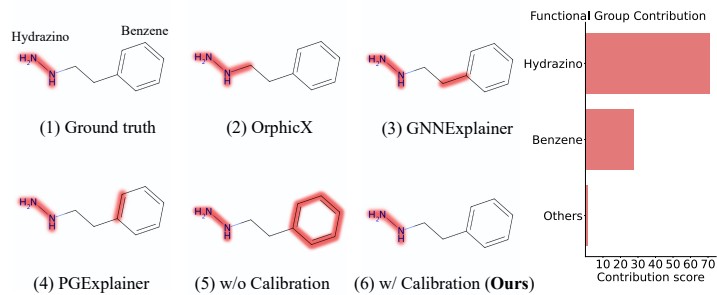

Figure 12: Explanation visualization of a molecule from the Liver dataset (left), and contribution scores of the identified functional groups offered by *MoleX* (right).

gression, as a more complicated linear model, achieves better performance compared to the simpler ones shown above. For more complex models, such as tree-based and ensemble learning models, the performance is even better, achieving incredible results across all seven datasets. These empirical studies suggest that augmenting statistical machine learning models with LLM knowledge significantly improves performance. Moreover, compared to simple models, the models exhibit more powerful data fitting capabilities become more predictive after the LLM augmentation. However, model complexity generally trades off with explainability. Considering this, we select the logistic regression as our base model due to its optimal balance between explainability and performance.

## A.13 CLASSIFICATION ANALYSIS VIA CONFUSION MATRIX

As shown in fig. 13, we visualize the classification result via confusion matrix at a random round on the Mutag and PTC-MR datasets. For Mutag, we achieve high precision in predicting the positive class due to fewer false positives and high recall for the positive class, reflecting the model's effectiveness in identifying positive instances. Furthermore, the model shows a good balance between precision and recall, with a low number of false positives and false negatives. For PTC-MR, the

Table 6: Classification Accuracy across different machine learning models over seven datasets (%)

| Method | Mutag | Mutagen | PTC-FR | PTC-FM | PTC-MR | PTC-MM | Liver |
|---|---|---|---|---|---|---|---|
| Ridge Regression | 90.7±1.2 | 84.1±1.3 | 72.4±2.0 | 65.2±2.0 | 69.8±1.4 | 77.5±1.5 | 58.1±1.6 |
| LASSO | 91.9±1.7 | 84.4±0.7 | 75.1±2.1 | 65.8±1.7 | 65.2±0.9 | 74.2±1.2 | 58.7±1.8 |
| Linear Discriminant Analysis | 89.9±1.9 | 83.6±1.2 | 75.2±1.9 | 65.7±1.9 | 69.3±1.8 | 76.8±2.0 | 57.7±1.3 |
| Polynomial Regression | 93.9±2.4 | 87.2±2.0 | 77.1±2.1 | 67.3±1.8 | 70.2±2.3 | 79.5±1.8 | 60.2±2.4 |
| Support Vector Machine | 93.9±1.6 | 86.6±1.5 | 73.4±1.9 | 69.3±2.6 | 69.5±2.0 | 78.6±1.3 | 61.5±2.9 |
| Decision Tree | 89.7±2.1 | 79.5±1.2 | 72.4±1.8 | 64.3±2.1 | 68.5±1.5 | 74.4±1.4 | 59.5±2.2 |
| Random Forest | 92.8±2.7 | 84.4±1.7 | 77.3±2.1 | 68.6±2.5 | 71.0±2.2 | 77.2±2.1 | 62.7±2.7 |
| Gradient Boosting Machine | 94.8±2.1 | 85.3±1.9 | 78.9±1.9 | 69.4±2.8 | 72.2±2.1 | 79.2±1.9 | 63.9±2.6 |
| XGBoost | 94.6±2.3 | 85.0±2.0 | 78.7±2.2 | 70.1±2.3 | 73.4±2.9 | 78.1±2.1 | 63.0±2.3 |
| MoleX (Ours) | 91.6±2.0 | 83.7±0.9 | 74.4±1.9 | 64.2±1.4 | 68.4±2.3 | 76.4±1.8 | 54.9±2.4 |

Table 7: Explanation Accuracy across different machine learning models over seven datasets (%)

| Method | Mutag | Mutagen | PTC-FR | PTC-FM | PTC-MR | PTC-MM | Liver |
|---|---|---|---|---|---|---|---|
| Ridge Regression | 92.8±1.1 | 89.5±1.3 | 79.0±1.2 | 78.1±1.6 | 72.5±2.5 | 69.7±2.3 | 82.4±1.7 |
| LASSO | 92.3±1.5 | 89.6±0.9 | 76.9±1.8 | 81.2±1.9 | 70.4±2.3 | 70.7±2.1 | 81.3±1.8 |
| Linear Discriminant Analysis | 92.9±1.8 | 88.5±1.9 | 80.7±2.3 | 80.1±2.2 | 71.7±2.8 | 71.3±1.6 | 87.8±1.6 |
| Polynomial Regression | 94.3±2.1 | 91.9±1.6 | 80.1±1.9 | 82.9±1.9 | 79.3±2.3 | 75.4±1.7 | 81.0±2.2 |
| Support Vector Machine | 92.0±1.7 | 92.0±1.6 | 84.7±2.2 | 86.3±2.0 | 80.1±2.3 | 76.0±2.3 | 81.9±2.1 |
| Decision Tree | 87.6±1.9 | 89.1±1.5 | 78.6±2.0 | 80.7±1.6 | 73.1±2.1 | 74.2±1.8 | 76.0±1.8 |
| Random Forest | 93.2±1.9 | 90.5±1.8 | 82.1±2.1 | 84.2±2.2 | 74.2±2.0 | 74.5±2.1 | 81.2±2.0 |
| Gradient Boosting Machine | 92.7±2.2 | 92.4±1.5 | 82.9±2.3 | 85.2±2.4 | 73.9±2.9 | 77.7±2.6 | 84.5±2.4 |
| XGBoost | 95.6±1.8 | 90.7±1.7 | 84.0±2.2 | 82.0±2.3 | 74.4±2.7 | 77.4±2.2 | 86.2±2.5 |
| MoleX (Ours) | 92.6±1.7 | 89.0±0.9 | 79.3±2.6 | 77.9±2.6 | 73.4±2.8 | 72.3±3.0 | 80.3±2.5 |

model achieves lower precision compared to the Mutag due to a higher number of false positives. The confusion matrix also suggests that the model struggles with false negatives and false positives, indicating areas for improvement. This analysis highlights the strengths and weaknesses of the model, providing insight for further model refinement.

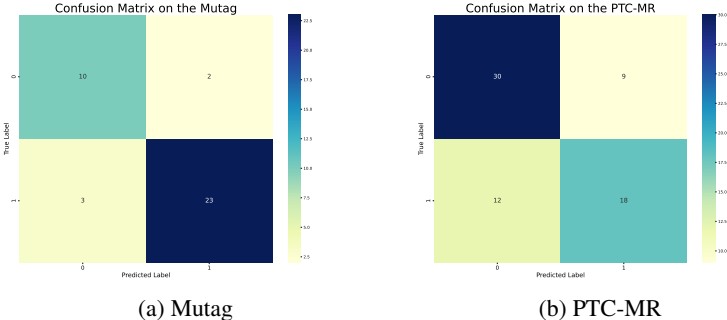

(a) Mutag          (b) PTC-MR

Figure 13: The confusion matrix of classification results on the Mutag and PTC-MR datasets

## A.14   AN ILLUSTRATION OF GROUP SELFIES

As illustrated in fig. 14, the 4-Nitroanisole ($C_7H_7NO_3$) can be represented by Group SELFIES with three functional groups separated by square brackets: a benzene ring, a nitro group, and a methoxy group (different functional groups are displayed in different colors).

## A.15   MORE EMPIRICAL EVALUATION ON THE ROBUSTNESS OF *MoleX*

As the molecular data is diverse, complex, and intrinsically noisy, we offer experiments on another three datasets, covering more extensive domains/tasks in molecular property prediction to demonstrate *MoleX*'s robustness. *MoleX* performs consistently excellent across all datasets and baselines,

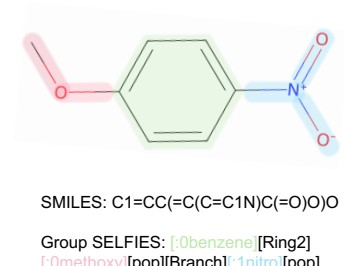

SMILES: C1=CC(=C(C=C1N)C(=O)O)O

Group SELFIES: [:0benzene][Ring2]
[:0methoxy][pop][Branch][:1nitro][pop]

Figure 14: Molecular representation of 4-Nitroanisole ($C_7H_7NO_3$)

Table 8: Classification accuracy over three datasets (%). The best results are highlighted in **bold**.

| Methods | BBBP | ClinTox | HIV |
|---|---|---|---|
| GCN (Kipf and Welling, 2016) | 78.5±0.8 | 78.2±1.0 | 72.1±0.8 |
| DGCNN (Zhang et al., 2018) | 80.0±0.9 | 79.0±1.1 | 73.2±1.4 |
| edGNN (Jaume et al., 2019) | 79.0±0.9 | 77.5±1.0 | 69.5±0.7 |
| GIN (Xu et al., 2018) | 82.0±0.7 | 80.9±0.9 | 74.0±1.3 |
| RW-GNN (Nikolentzos and Vazirgiannis, 2020) | 81.0±1.0 | 78.5±1.0 | 75.5±0.4 |
| DropGNN (Papp et al., 2021) | 83.0±0.9 | 81.0±0.8 | 64.5±0.6 |
| IEGN (Maron et al., 2018) | 85.5±1.0 | 80.1±0.5 | 76.0±0.9 |
| LLAMA3.1-8b (Dubey et al., 2024) | 69.0±2.5 | 52.0±2.7 | 56.0±1.5 |
| GPT-4o (Achiam et al., 2023) | 74.5±2.3 | 56.4±2.5 | 64.5±1.8 |
| ChemBERTa-2 (Ahmad et al., 2022) | 78.0±1.5 | 71.5±1.4 | 73.0±0.6 |
| Logistic Regression | 66.5±0.8 | 60.2±0.6 | 60.1±0.7 |
| Decision Tree (Quinlan, 1986) | 70.3±0.8 | 62.8±0.6 | 66.2±0.8 |
| Random Forest (Breiman, 2001) | 73.5±0.9 | 68.5±0.7 | 69.8±1.9 |
| XGBoost (Chen and Guestrin, 2016) | 74.2±0.8 | 67.8±0.8 | 70.2±1.2 |
| w/o Calibration | 80.6±1.3 | 85.9±0.7 | 75.6±1.3 |
| **w/ Calibration (Ours)** | **93.1±0.6** | **94.1±0.8** | **81.3±1.4** |

showcasing its effective generalizability. The results of classification and explanation accuracy are shown in table 8 and table 9, respectively.

## A.16 BROADER IMPACT

This study on explainable molecular property prediction using an LLM-augmented linear model offers significant real-world applications. The efficiency of linear models enables fast inference on large-scale molecular data, potentially accelerating drug discovery and materials design. Enhanced by LLM-derived features, our method combines predictive accuracy, cost-effectiveness, and computational efficiency, addressing critical needs in fields like healthcare and materials science. Its high explanation accuracy provides faithful insights into structure-property relationships, fostering adoption in high-stakes domains and supporting scientific discovery. Additionally, this balance of accuracy, explainability, and efficiency serves as a template for developing trustworthy AI in other fields, with potential impacts on personalized medicine and sustainable chemistry. However, responsible implementation is crucial to mitigate risks, such as over-reliance on predictions or misuse in harmful molecule design, emphasizing the need for expert validation and research into limitations.

## A.17 LIMITATIONS AND FUTURE WORKS

The proposed explainable molecular property prediction method has some limitations and needs further studies.

- **Generalizability:** Enhancing the generalizability of explainable models to deal with different molecular datasets across various chemical domains while preserving explainability to structure-property relationships remains a persistent challenge.
- **Impact of LLM choices:** Though our empirical studies discuss the model performance of Llama3.1 and GPT-4o on molecular property prediction, LLM quality is still a topic that

Table 9: Explanation accuracy over three datasets (%). The best results are highlighted in **bold**.

| Methods | BBBP | ClinTox | HIV |
|---|---|---|---|
| GCN (Kipf and Welling, 2016) | 75.1±0.4 | 74.6±0.6 | 67.6±0.6 |
| DGCNN (Zhang et al., 2018) | 77.6±1.1 | 79.2±0.5 | 73.8±1.1 |
| edGNN (Jaume et al., 2019) | 78.9±0.2 | 74.8±0.2 | 71.6±0.6 |
| GIN (Xu et al., 2018) | 80.4±0.7 | 77.1±0.8 | 70.3±0.8 |
| RW-GNN (Nikolentzos and Vazirgiannis, 2020) | 79.5±0.4 | 69.4±0.6 | 69.5±0.7 |
| DropGNN (Papp et al., 2021) | 72.6±0.6 | 76.7±0.2 | 74.4±0.3 |
| IEGN (Maron et al., 2018) | 80.8±0.7 | 79.1±0.4 | 69.5±1.2 |
| Logistic Regression | 67.9±0.3 | 61.9±0.2 | 61.8±0.6 |
| Decision Tree (Quinlan, 1986) | 68.4±1.5 | 66.8±0.8 | 64.0±1.2 |
| Random Forest (Breiman, 2001) | 73.3±1.1 | 68.3±1.7 | 65.7±1.3 |
| XGBoost (Chen and Guestrin, 2016) | 73.5±1.4 | 65.5±1.6 | 67.8±0.9 |
| w/o Calibration | 81.1±1.8 | 78.6±1.5 | 71.2±1.1 |
| **w/ Calibration (Ours)** | **90.8±1.6** | **92.8±1.9** | **82.4±1.2** |

deserves to be explored in-depth. Future studies may discuss how LLM choices impact the augmented linear model, e.g., model performance change using weak LLMs or LLMs without fine-tuning.

- **Trade-off between complexity and performance:** In pursuit of explainability, we employ a linear model, which inherently risks underfitting when faced with complex data patterns. Our preliminary experiments comparing *MoleX* with more sophisticated statistical learning models show marginally better performance from these complex models. Future research could explore the trade-off between model complexity and performance in the context of LLM knowledge augmentation and investigate optimal balances between explainability and performance.

