# OpenReview forum: "Unveiling Molecular Secrets: An LLM-Augmented Linear Model for Explainable and Calibratable Molecular Property Prediction"
_ICLR.cc/2025/Conference — Submitted to ICLR 2025_

### Official Review · Reviewer_CoqY · 2024-11-02

**Soundness:** 2
**Presentation:** 1
**Contribution:** 2
**Rating:** 5
**Confidence:** 2

**Summary:**

The authors are interested in learning a molecular property prediction model using a large language model (LLM) with explainability. The proposed framework called MoleX consists of four components.

First, it extracts a useful feature from LLM embedding using the variational information bottleneck (VIB) principle (Section 4.1).
Second, it applies dimensionality reduction to the feature extracted in the first step based on the explainable functional principal component analysis, which roughly corresponds to PCA with sparsity-inducing penalty to each component (Section 4.2).
Third, using the explainable feature obtained in the second component, it fits a linear model that maps the explainable feature to the target variable (Section 4.2).
Fourth, with the predictive model fixed, it fits another linear model using the features not used in the third component for residual calibration (Section 4.2).

The authors evaluate the effectiveness of MoleX in terms of its predictive power as well as its explainability and computational efficiency.
The experimental results suggest MoleX is better than baseline methods in every aspect.
Sensitivity analyses on hyperparameters are conducted in Section 5.3, but most of the details are sent to the appendix.

**Strengths:**

- The problem setting to seek for a predictive model for molecules with explainability is a high-demand setting for practitioners, and I would appreciate the authors for achieving the state-of-the-art performance on this problem setting.
- The experiments are exhaustive in terms of the methods compared and experiments conducted, although many of the details are not in the main part of the paper.

**Weaknesses:**

- The overview of MoleX is not clear, especially, how we train it and use it for obtaining prediction with explanations in a step-by-step manner. Although Figure 1 suggests a vague overview of it, as far as I am aware of, there is no rigorous explanation of it. Since it is very essential to describe MoleX, I would like to request the authors to explain it in an algorithmic way in the main part of the paper (not in the appendix).
- There are many mathematically incorrect/unclear descriptions, which pose challenges in understanding the paper.  A few examples are as follows:
  - Line 152: Since $g$ and $y$ are not sets, $f\colon g\to y$ should be $f\colon g\mapsto y$.
  - Definition 4.1: There are multiple concepts that are not explained, namely, an operator $\mathcal{D}^2$ and an inner product $\langle \cdot, \cdot \rangle_{\gamma}$. $k$ appears both in the objective function and constraints, but there is no explanation about it.
  - Line 260: $\ell_0$ penalty does not directly appear in Def. 4.1 (in a finite dimensional case, the regularization term boils down to the $\ell_0$ penalty, but the definition is given for the infinite dimensional case).
  - Lines 309-311: Given that $f_H(x)\in\mathbb{R}^{d_c}$ and $f_R(x)\in\mathbb{R}^{d_r}$, $\langle f_H(x), f_R(x) \rangle$ cannot be defined in a natural way.

**Questions:**

- Please add an algorithm describing the whole procedure of MoleX including both its training and inference.
- I still don't understand why the explanation accuracy of the proposed method improves when with calibration. Could it be because the dimensionality reduction was too aggressive that a part of the features necessary for explanation was removed? I would appreciate it if the authors could point out any misunderstanding of mine if any / provide explanation on it.

---

> ### Author Response · Authors · 2024-11-23
>
> We sincerely appreciate your constructive feedback and your insightful concerns.  We are glad that you find our study **"state-of-the-art"** and **"exhaustive experiments"**. Below we will address your questions in detail.
>
> ---
>
> **W1, Q1: Add an algorithm to explain the workflow**
>
> We appreciate your suggestion to use an algorithm to explain the workflow. We add pseudo code describing the procedure of the LLM fine-tuning and the training and inference of *MoleX* in the Appendix. Due to the limited space and the length of the algorithm, we apologize that it is difficult to include them in the main part of the paper. Considering its importance in facilitating readers' understanding of *MoleX*'s mechanism, we have included it as the first part of the Appendix. The manuscript has been updated accordingly.
>
> ---
>
> **W2: Revision to math descriptors.**
>
> We sincerely apologize for the misused or unclear math descriptors as we explained some of them in the Appendix. Thank you for pointing them out. We have fixed them with detailed explanations and checked our manuscript over and over again. We now updated our manuscript.
>
> ---
>
> **Q2: The reason why residual calibration improves the explanation accuracy.**
>
> This is a very good question touching the core of our design. We introduce the residual calibration to compensates for the expressiveness gap between linear models and complex LLM embeddings. Here, the calibrator is designed to fix the prediction errors stemming from the linear model's underfitting limitation.
>
> Note that the Group SELFIES, represented by functional groups, serves as the input data. Residual calibration enables *MoleX* to iteratively correct mispredicted functional groups and their interactions, aligning explanations with chemical expertise. By progressively calibrating prediction errors toward the ground truth, the method ensures explanations use chemically accurate functional groups and interactions. This process enhances both the accuracy (in terms of AUC) and the faithfulness (in terms of chemical expertise) of explanations.

---

> ### Comment · Reviewer_CoqY · 2024-11-24
> **Re: Official Comment by Authors**
>
> Thank you very much for the response.
> After reading the response and the manuscript again, I could not understand the definitions of $f_H$ and $f_R$.
> - In the pseudo code added by the authors, these are given without any explanations.
> - Line 288 says $f_H$ is defined in Equation 3.1, but it defines the objective function to learn $r$.
> I would like to ask the authors to clarify the definitions.
>
> For the pseudo code, since it is the only explanation on the overall process of the proposed method, I consider it is an essential part of the paper that should be included in the main body.
> ["Call for papers"](https://iclr.cc/Conferences/2025/CallForPapers) says that,
>
> > Authors may use as many pages of appendices (after the bibliography) as they wish, but reviewers are not required to read the appendix.
>
> and I understand it is mandatory to put all the essential materials in the main body, not in Appendix (otherwise, the page limit is meaningless). Therefore, I would like to request the authors to revise the paper so that the main body is self-contained.

---

> ### Author Response · Authors · 2024-11-25
>
> We thank your prompt feedback. Your questions are addressed as follows.
>
>  1. $f_H(x)$ and $f_R(x)$ are dimensionality-reduced features used to train the explainable model $h$ and residual calibrator $r$, respectively. $f_H(x)$ captures explainable features related to $y$, while $f_R(x)$ captures the remaining features with residual variance. As the orthogonality condition given, they satisfy $\langle f_H(x), f_R(x) \rangle = 0$. In the molecular setup, $f_H(x)$ represents functional group features critical to $y$, fitting by $h$ to explain their relationship, while $f_R(x)$ captures residual variance not explained by $h$. **We have added explanations into our pseudo code**.
>
> 2. Apologies for the lack of clarity. We have corrected the $r$ notation in Equation 3.1, as both $h$ and $r$ are learned using this objective. In our implementation, the model first learns $h$, followed by $r$ (which learns residuals of $h$). The learning of $r$ involves residual calibration, updating the parameters of both $h$ and $r$ to improve overall model performance.
>
> **We have revised the paper and included the pseudo code in the main body. Modifications are highlighted in blue in the updated manuscript. We appreciate your invaluable suggestions again.**

---

### Official Review · Reviewer_s7A7 · 2024-11-03

**Soundness:** 2
**Presentation:** 1
**Contribution:** 1
**Rating:** 3
**Confidence:** 4

**Summary:**

The paper introduces MoleX, a framework combining linear models with large language model (LLM) embeddings for explainable and accurate molecular property prediction. The framework's unique features include:
- Information bottleneck-inspired fine-tuning and sparsity-inducing dimensionality reduction to enhance the informativeness of LLM embeddings.
- Residual calibration to address prediction errors, thereby boosting the performance of the linear model without sacrificing explainability.
- Quantification of functional group contributions to molecular properties through n-gram coefficients.

**Strengths:**

- Experimental results across seven datasets demonstrate that MoleX outperforms existing baselines (both GNN and LLM models) in predictive and explanation accuracy.
- The evaluation metrics, including classification accuracy, explanation AUC, and computational efficiency, present a holistic view of the model's advantages. The ablation studies reinforce the robustness of the method by exploring different parameter choices and model components.

**Weaknesses:**

- The methodology section, while comprehensive, largely restates existing techniques without significant novel contributions. Many of the core ideas, such as the information bottleneck fine-tuning and dimensionality reduction via EFPCA, are established approaches adapted with minimal modification. This raises concerns regarding the originality of the proposed framework.
- The claims surrounding the methodological innovations appear overstated, given that the fundamental components, such as fine-tuning with variational information bottleneck and principal component analysis, are widely known and previously employed in similar contexts. The application of these techniques, while effective, may not sufficiently differentiate this work as a significant leap forward in the field.

**Questions:**

- I think the authors would want to reformulate the problem and reorganize the paper to clearly and precisely state their contributions and claims, and describe their method framework, better following academic integrity.

---

> ### Author Response · Authors · 2024-11-23
>
> We sincerely appreciate your constructive feedback and your insightful concerns.  We are glad that you find our study **"outperforms existing baselines"** and **"holistic experiment evaluation"**. Below we will address your questions in detail.
>
> ---
>
> Since the mentioned W1, W2, and Q pose the same concern, we would clarify it here. In summary, our technical innovation and contributions involve:
>
> 1. **Innovative adaptation.**
>    We effectively adapt/refine and integrate the established approaches to achieve state-of-the-art explanation and predictive accuracy, even compared with advanced neural nets and LLMs.
>    *(as commented by reviewer P2P1)*
>
> 2. **Novel design of residual calibration.**
>    We introduce a novel residual calibration module that compensates for the expressiveness gap between linear models and complex LLM embeddings, effectively restoring predictive performance while preserving explainability through iterative prediction error recapturing.
>    *(as commented by reviewers 3Cvr, P2P1, and CoqY)*
>
> 3. **A leap forward in explainable molecular property prediction.**
>    Linear models offer high explainability for their simple architecture but struggle with challenging tasks. However, we design novel approaches to augment them with LLM knowledge and residual calibration, enabling them to handle complex molecular data. Our strong performance marks a significant advancement in explainable molecular property prediction.
>    *(as commented by reviewers 3Cvr and P2P1)*
>
> 4. **Chemical expertise-aligned explanations.**
>    We explain molecular properties with chemically meaningful functional groups and their interactions, highly aligning with chemical domain knowledge.
>    *(as commented by reviewers wPWb and CoqY)*
>
> 5. **Holistic analysis.**
>    We provide exhaustive theoretical and empirical justification to demonstrate our framework.
>    *(as commented by reviewers 3Cvr and CoqY)*
>
> ---
>
> While individual components such as linear models and IB are established, our key innovation consists of adapting and integrating them to create an explainable linear model enhanced by LLM embeddings for molecular property prediction. For instance, our adaptation of the variational information bottleneck is uniquely tailored to extract task-relevant knowledge during LLM embeddings in molecular contexts, which has not been previously explored. Additionally, we propose EFPCA to effectively extract the most relevant information from the complex LLM embedding space.
>
> The novelty of this work has also been acknowledged by other reviewers. We received comments such as **"stands out for its innovative approach/innovative"** *(reviewers P2P1 and wPWb)*, and **"creative solution"** *(reviewer P2P1)*.

---

> > ### Comment · Reviewer_s7A7 · 2024-11-27
> > **Discussion**
> >
> > I don't think this response clearly addressed my concerns. What I meant is you shouldn't overstate things by writing your methodology in a way that highly correlates with existing literature without proper discussion with them. In another word, the theory/methods you reused/redefined/reformulated from other works should be clearly defined as part of preliminaries.
> >
> > Also, I think most of the methods introduced in this paper are highly orthogonal to molecular property prediction, even LLMs. They are very general that is on top of any embedding model. If you really want to emphasize these, you probably need to rewrite the paper with claims built off embedding models rather than "LLMs" and "molecular property prediction".
> >
> > Lastly, please properly address the reviewer's concerns by your useful evidence, not by citing other reviewer's comments. This is the discussion period between the authors and the reviewer. There is another discussion stage afterward between reviewers and ACs. Citing comments from other reviewers wouldn't be of any help of making your response stronger.

---

> > > ### Author Response · Authors · 2024-11-28
> > >
> > > Thank you for your insightful feedback. We apologize for not fully addressing your concerns in the previous response. In our revised manuscript, we have clearly introduced these methods and thoroughly discussed their adaptation/refinement within our framework.
> > >
> > > Our primary contribution lies in the novel integration and adaptation of these techniques specifically for molecular property prediction leveraging LLM knowledge. While the individual methods are foundational, our tailored refinement—augmenting a simple linear model with LLM knowledge to enhance chemistry-aligned explainability in the molecular context—represents a technical innovation. We have updated the manuscript (please check page 3, 4, 5, 6 where we highlighted the modifications in blue) to emphasize how our framework leverages LLMs to address challenges unique and novel to molecular data within the linear model architecture, distinguishing our work from prior studies.
> > >
> > > Regarding your concern that the methods introduced are orthogonal to molecular property prediction and LLMs, we clarify that our framework is specifically designed to address challenges in this domain. The use of Group SELFIES for molecular representation and the fine-tuning of pre-trained LLMs to maximize informativeness and capture chemical semantics are novel contributions. Our residual calibration module addresses the expressiveness limitations of linear models when handling complex molecular embeddings from LLMs—an issue particularly relevant to molecular data. It is innovatively designed to enhance both explainability and performance simultaneously. While some components (e.g., linear models, dimensionality reduction) are established techniques, their application and integration in our framework are specialized for enhancing both explainability (as our explainability arises from domain-expertise-aligned explanations) and performance in molecular property prediction. Thus, we believe our work offers a meaningful, domain-specific contribution, not a generic approach applicable to any embedding model.
> > >
> > > We hope the provided clarifications and explanations, except for referring other reviewers’ comments, can better resolve your concerns. Thank you again for your feedback. We remain committed to improving our work and address your concerns.

---

### Official Review · Reviewer_wPWb · 2024-11-03

**Soundness:** 3
**Presentation:** 1
**Contribution:** 2
**Rating:** 3
**Confidence:** 5

**Summary:**

This paper introduces MoleX, a framework for explainable molecular property prediction that combines large language model (LLM) embeddings with a linear model to achieve both accuracy and interpretability. MoleX leverages fine-tuned LLM knowledge and residual calibration to enhance prediction accuracy while enabling CPU-efficient inference, showing up to 12.7% improved performance and significant parameter reduction compared to standard LLMs.

**Strengths:**

1. **Reduced Complexity via Linear Model**: The use of a linear model effectively lowers overall computational complexity, making the framework suitable for large-scale applications.

2. **Efficient Dimensionality Reduction**: The dimensionality reduction technique minimizes computational demands, optimizing the handling of high-dimensional LLM embeddings.

3. **High Interpretability**: The model offers strong interpretability, providing reliable, chemically meaningful explanations for molecular predictions.

**Weaknesses:**

This paper, though innovative, exhibits certain weaknesses:

1. **Lack of Comparison with Relevant Explainability Studies**: The authors did not compare their approach with other information bottleneck based methods, for example, SGIB and VGIB [1,2], which also employ IB techniques to identify explainable subgraphs most relevant to molecular properties. Including these comparisons would provide a more comprehensive evaluation of the proposed method.

2. **Poor Presentation**: The presentation of the paper is lacking in clarity, with several areas insufficiently explained. This detracts from the paper’s readability and hinders a full understanding of the methodology. This is my primary reason for recommending rejection, as the paper requires major revision to improve clarity. Specific issues are detailed in the "Questions" section.

3. **Omission of LLM-Based Method Comparisons in Table 2**: In Table 2, the authors do not compare LLM-based methods evaluated in Table 1. The reasons for this omission should be explicitly addressed within the paper to clarify the experimental choices and maintain consistency in the evaluation.

[1] Yu, Junchi, et al. "Graph Information Bottleneck for Subgraph Recognition." International Conference on Learning Representations.

[2] Yu, Junchi, Jie Cao, and Ran He. "Improving subgraph recognition with variational graph information bottleneck." Proceedings of the IEEE/CVF Conference on Computer Vision and Pattern Recognition. 2022.

**Questions:**

1. **Line 261**: How are the $\ell_0$ penalty, $a_{kj}$, and $\phi_j(t)$ derived? These symbols should be explained in the main text rather than in the appendix to ensure readers understand their significance without having to refer to supplementary materials.

2. **Input Consistency for Explainable Model and Residual Calibrator**: Are the inputs for the explainable model and residual calibrator identical, specifically the reduced LLM embeddings after EFPCA? Clarifying this point would help readers better understand the workflow and the rationale behind the model architecture.

3. **Line 308**: How is $f$ defined, and are $f_H$ and $f_R$ the outputs of EFPCA? If so, why are they different, and how does EFPCA produce distinct outputs from the same input? Detailed clarification on this process would enhance the comprehension of the model’s internal mechanisms.

4. **Inconsistent Notation**: The notation varies across different sections and should be made consistent to avoid confusion.

---

> ### Author Response · Authors · 2024-11-23
>
> We sincerely appreciate your constructive feedback and your insightful concerns. We are glad that you find our study **"high interpretability"**, **"innovative"**, and **"efficient, reduced complexity"**. Below we will address your questions in detail.
>
> ---
>
> - **W1: Lack of Comparison with Relevant Explainability Studies.**
>
>   Thanks for your suggestions. However, our objective is to maximize task-relevant information during LLM fine-tuning, which is realized with information bottleneck. In contrast, SGIB and VGIB are proposed to identify explainable subgraphs in GNNs via information bottleneck. The purpose of using IB in *MoleX* differs from the mentioned papers, and comparing *MoleX* with SGIB and VGIB might deviate from the main idea of this study.
>
> ---
>
> - **W2, Q1-4: Mathematical Explanation.**
>
>   We apologize for missing these details. Since Q1-4 specify the confusion in W2, we would elaborate our explanations to these questions here. Your questions are addressed as follows.
>
>   - **Line 261. More Explanations**
>
>     To explain how the $\ell_0$ penalty, the coefficients $a_{kj}$, and the basis functions $\phi_j(t)$ are derived in the EFPCA, we explain each component respectively:
>
>     1. **The basis functions $\phi_j(t)$**
>
>        The basis functions $\phi_j(t)$ are derived by selecting a set of functions that have local support on subintervals $S_j \subset [a, b]$. Typically, we use B-spline basis functions because they are piecewise polynomial functions that are nonzero only within specific intervals. This local support property enables the principal components $\xi_k(t)$ to capture localized features in the data, which enhances explainability.
>
>     2. **The coefficients $a_{kj}$**
>
>        The coefficients $a_{kj}$ are derived through the expansion of the principal components $\xi_k(t)$ in terms of the chosen basis functions. We express $\xi_k(t)$ as a linear combination $\xi_k(t) = \sum_{j=1}^p a_{kj} \phi_j(t)$ where $p$ is the number of basis functions. The coefficients $a_{kj}$ are determined by solving the optimization problem posed in the EFPCA, which aims to maximize the variance explained by $\xi_k(t)$ while inducing sparsity.
>
>     3. **The $\ell_0$ penalty $\rho_k \|| a_k \||_0$**
>
>        The $\ell_0$ penalty **$\rho_k \|| a_k \||_0$** is derived from the need to encourage sparsity in the coefficients $a_{kj}$. Here, **$ \|| a_k \||_0$** counts the number of nonzero coefficients in principal components By including the $\ell_0$ penalty into our designed optimization objective, we can impose a penalty proportional to the number of nonzero coefficients. This promotes solutions where many $a_{kj}$ are exactly zero, resulting in principal components that are sparse and focused on the most significant features of the data.
>
>     Consequently, the derivation involves:
>
>     - Selecting basis functions $\phi_j(t)$ with local support to enable explainability.
>     - Expressing $\xi_k(t)$ as a linear combination of these basis functions and deriving the coefficients $a_{kj}$ through optimization.
>     - Introducing the $\ell_0$ penalty to induce sparsity in $a_{kj}$, which enhances the explainability of the resulting principal components.
>
>   - **Input Consistency**
>
>     Great question. The explainable model and residual calibrator share the same input: dimensionality-reduced LLM embeddings. *MoleX*'s workflow involves three steps:
>
>     1. Feeding the input into the explainable linear model and collecting prediction error samples.
>     2. Inputting the prediction errors into the residual calibrator (also a linear model).
>     3. Aggregating the calibrator's output with the explainable linear model.
>
>     During its workflow, the calibrator recaptures prediction errors and simultaneously updates the entire model's parameters, ensuring consistent input throughout the process.

---

> ### Author Response · Authors · 2024-11-23
>
> ***continue***
>
> - **Line 308**
>
>     **Are $\boldsymbol{f_R}$ and $\boldsymbol{f_H}$ outputs of EFPCA and why are they different?**
>
>     Thanks for your question. Yes, the $f_H$ and $f_R$ are outputs of EFPCA (as definitions are given in the preliminaries section), and they are different because EFPCA applies dimensionality reduction to the feature vector $f(x)$, transforming it into a set of orthogonal principal components (PCs). We construct $f_H(x)$ by selecting the top PCs that are explainable with respect to the molecular property $y$, while $f_R(x)$ consists of the remaining principal components capturing residual variance. Since these PCs are orthogonal, $f_H(x)$ and $f_R(x)$ are distinct and satisfy $\langle f_H(x), f_R(x) \rangle = 0$, resulting in EFPCA producing different outputs from the same input $f(x)$. Specifically, the vector $f_H(x)$ contains features corresponding to functional groups that contribute significantly to $y$. These features are used by the explainable model $h$ to interpret the relationship between functional groups and $y$. The vector $f_R(x)$ comprises the remaining features, representing additional variance in $y$ not captured by $h$.
>
>     **How does EFPCA produce distinct outputs from the same input?**
>
>     This is an inspiring question. Note that EFPCA ensures that all principal components are orthogonal ($\langle \text{PC}_i, \text{PC}_j \rangle = 0$ for $i \ne j$), so $f_H(x)$ and $f_R(x)$ are orthogonal, satisfying $\langle f_H(x), f_R(x) \rangle = 0$. This orthogonality allows $h$ and $r$ to operate on different subsets of the transformed functional group features without interfering with each other. Thus, $f_H(x)$ and $f_R(x)$ are distinct both in terms of their feature content and their roles in modeling $y$, enhancing model performance without compromising explainability.
>
>     Since $f_H(x)$ and $f_R(x)$ are different components of $f(x)$, EFPCA produces distinct outputs from the same input $f(x)$ by decomposing the feature space into these orthogonal subspaces. By projecting $f(x)$ onto the subspace spanned by the selected PCs, we obtain $f_H(x)$, and projecting onto the remaining orthogonal subspace yields $f_R(x)$. Therefore, EFPCA generates $f_H(x)$ and $f_R(x)$ as distinct outputs corresponding to explainable and residual features, respectively.
>
>   - **Inconsistent Notation**
>
>     We appreciate your advice. We have offered more details and explanations to unclear or non-explained math notations in our updated manuscript.
>
> ---
>
> - **W3: Omission of LLM-Based Method Comparisons in Table 2**
>
>   We sincerely thank your reminders and will clarify this in our updated manuscript. We did not offer consistent evaluation metrics in Table 2 for LLMs because AUC, a widely-used metric for explanation accuracy, is inappropriate for LLMs' continuous, structured text outputs: LLMs' probabilistic distributions over large vocabularies are incompatible with the binary classification framework required for AUC computation.
>
> ---

---

> ### Comment · Reviewer_wPWb · 2024-11-26
> **Need to improve the writing.**
>
> The definition of $\|a_k\|$ is clear to me. However, every symbol appearing in the main text must be explicitly defined within the main text itself, rather than relying on definitions provided in the appendix. Additionally, the definition for EFPCA is directly adopted from [1], but it is inconsistent with the notational framework used in this paper. Ensuring consistent notation throughout the paper is essential for clarity and readability. For instance, the relationship between $(f_H, f_R)$, and $\xi$ in EFPCA should be explicitly clarified when $f_H$ and $f_R$ first appear at Line 172. Alternatively, you could specify that these elements will be discussed in detail in specific sections (e.g., Section XX and XX).
>
> The definition of $t$ also presents issues of clarity and consistency. In EFPCA, $t$ represents the independent variable within a compact interval $[a, b]$. However, in Section 4.1 of this paper, $t$ is firstly defined as LLM embeddings, while in the algorithm section, these embeddings are instead denoted as $x^{\text{emb}}$. Moreover, the notation for the $i$-th group SELFIES is inconsistent: it is defined as $x^{(i)}$ at Line 168 but appears as $x_i$ in Algorithm 1.
>
> Such inconsistencies in notation and definitions detract from the overall coherence and readability of the paper. The paper currently feels fragmented, resembling a collection of disjointed sections rather than a cohesive, self-contained work. A major revision is still needed to improve the overall presentation, ensure notational consistency, and enhance the clarity of the paper.
>
> [1] Zhenhua Lin, Liangliang Wang, and Jiguo Cao. Interpretable functional principal component analysis. Biometrics, 72(3):846–854, 2016.
>
> BTW, I am satisfied with the feedback for W1 and W3.

---

> > ### Author Response · Authors · 2024-11-28
> >
> > We sincerely appreciate your recognition and suggestions. We have updated the manuscript to resolve issues including but not limited to presentation /notational unclarity in Algorithm 1, preliminaries, and 4.1 fine-tuning.
> >
> > The modular structure of our work arises from the challenges we faced in extracting molecular representations with distinct semantic information from LLMs, adapting them to our linear model framework, and improving chemical expertise-aligned explainability in the context of molecular science, etc.. Each module was designed to address a specific aspect of these challenges, which may have potentially contributed to the perception of disjointedness. We have made effort to better clarify them in our revised manuscript.
> >
> > In the updated manuscript, we worked on detailing these modules into a cohesive narrative and explaining how each component contributes to the goal of explainable molecular property prediction. Additionally, we ensure notational consistency and enhance clarity throughout the paper to provide a more polished presentation. Thank you again for raising this important point and we are committed to addressing it in our revisions.

---

### Official Review · Reviewer_P2P1 · 2024-11-04

**Soundness:** 3
**Presentation:** 3
**Contribution:** 3
**Rating:** 6
**Confidence:** 4

**Summary:**

The paper introduces MoleX, a framework designed to achieve accurate and interpretable molecular property predictions by leveraging the knowledge embedded in large language model (LLM) embeddings while maintaining a simple, interpretable linear model structure. MoleX combines the predictive power of LLM embeddings with the transparency of linear models, aiming to bridge the gap between complex models that excel in prediction and simpler models that offer more interpretability.

**Strengths:**

The paper stands out for its innovative approach to bridging the gap between prediction accuracy and interpretability in molecular property prediction. While explainability and accuracy have traditionally been seen as trade-offs, MoleX proposes a hybrid framework that leverages the strengths of both linear models and LLMs. This combination—using LLM embeddings to build a more interpretable linear model, while employing information bottleneck fine-tuning and sparsity to adapt these embeddings—is a creative solution to a long-standing problem. Furthermore, the introduction of a residual calibration strategy to enhance a linear model’s capacity in representing complex LLM-derived features demonstrates originality by addressing a specific limitation of linear methods. Overall, this paper is a meaningful contribution that combines originality with technical rigor, clarity, and significant practical implications.

**Weaknesses:**

While MoleX offers a creative use of LLM embeddings for interpretable molecular property prediction, the novelty might be limited as the work relies heavily on existing LLM representations and linear models. The use of embeddings from pre-trained LLMs combined with linear regression isn’t entirely new, as similar approaches have been applied in other fields for simplifying complex models while preserving interpretability. The authors could enhance the novelty by providing a clearer comparison of MoleX with similar hybrid approaches in other domains, or by further distinguishing their method’s unique aspects in the context of molecular property prediction. The authors can compare their method with prototype-based approaches as ProtoPNet [1] and xDNN [2].

[1] Chen, Chaofan, et al. "This looks like that: deep learning for interpretable image recognition." Advances in neural information processing systems 32 (2019).

[2] Angelov, Plamen, and Eduardo Soares. "Towards explainable deep neural networks (xDNN)." Neural Networks 130 (2020): 185-194.

**Questions:**

1. How is “faithfulness” in MoleX’s explanations quantified or evaluated? Could you provide more details on the metrics or assessment criteria used to determine that the model’s explanations are accurate and reliable from a chemical standpoint?

2. How does MoleX differ from or improve upon prototype-based approaches, which are also commonly used to enhance interpretability by referencing representative examples?

3. Could you provide more details about the residual calibration strategy? Specifically, how is it implemented, what types of prediction errors does it address, and how does it affect the interpretability of the final predictions?

4. Since MoleX relies on a linear model, are there cases where it struggles to represent the complexity of molecular data, even with LLM embeddings and residual calibration? How do you manage this limitation?

5. Which metrics were used to evaluate the explainability of MoleX compared to other interpretable models?

---

> ### Author Response · Authors · 2024-11-23
>
> We sincerely appreciate your constructive feedback and your insightful concerns.  We are glad that you find our study **"meaningful contribution"**, **"innovative"**, and **"technical rigor, clarity, and significant practical implications"**. Below we will address your questions in detail.
>
> ---
>
> - **W, Q2: Difference of *MoleX* between prototype-based approaches and our advantages.**
>
>   We appreciate your concerns about the difference among explainability approaches. *MoleX* differs from prototype-based approaches, such as ProtoPNet and xDNN, in several aspects and holds some advantages over them.
>
>   - **Global vs. local explanations.** *MoleX* provides global explanations by modeling the data distribution over the entire feature space, while prototype-based approaches offer case-based, post-hoc explanations. Global explainability better reveals the model's overall behavior, making *MoleX* more capable of interpreting the entire variability in complex molecular structures.
>
>   - **Explicit explainability.** *MoleX*'s explainability arises from the linearity of the model's coefficients, where individual features' contributions provide explanations. In contrast, prototype-based methods rely on similarity to real examples without explicitly articulating the key contributing features, making the precise reasoning harder for humans to understand. Thus, *MoleX* offers more explicit explanations behind predictions.
>
>   - **Chemical expertise-aligned explanations.** In the context of our study, we explain the molecular properties with chemically meaningful functional groups and their interactions, highly aligning with chemical expertise.
>
>   - **Innovative design to enhance explainability.** We introduce a residual calibration strategy that improves both the faithfulness and accuracy of explainability via iterative prediction error fixing.
>
> ---
>
> - **Q1: The evaluation of faithfulness from a chemical standpoint.**
>
>   This is a good question. We evaluate how faithful the explanations offered by *MoleX* are through chemical expertise to highlight its explainability (i.e., how correct the explanations are from a chemical perspective). As mentioned in the Appendix, domain experts marked correct functional groups that influence molecular properties as ground truth. Empirically, *MoleX* offers explanations highly aligned with these ground truths.
>
> ---
>
> - **Q3: More details about residual calibration strategy.**
>
>   These are very great questions. We are glad to summarize more details about residual calibration to resolve your confusion.
>
>   - **High-level ideas.** We introduce residual calibration to address the linear model's limited expressiveness with high-dimensional LLM embeddings. It refits prediction errors caused by underfitting, with a training objective designed to improve overall performance by iteratively calibrating the explainable model's residuals (i.e., update both models' parameters during prediction error calibration).
>
>   - **How residual calibration improves explainability.** Note that the Group SELFIES, represented by functional groups, serves as the input data. Residual calibration enables *MoleX* to iteratively correct mispredicted functional groups and their interactions, aligning explanations with chemical expertise. By progressively calibrating prediction errors toward the ground truth, the method ensures explanations use chemically accurate functional groups and interactions. This process enhances both the accuracy (in terms of AUC) and the faithfulness (in terms of chemical expertise) of the explanations.
>
>   - **Implementation details.** The workflow mainly involves three steps:
>     1. Feeding the input into the explainable linear model and collecting residual samples.
>     2. Feeding residual samples into the residual calibrator (also a linear model).
>     3. Aggregating the output of both models. The training objective updates both models' parameters simultaneously while fixing mispredicted samples.
>
> ---

---

> > ### Author Response · Authors · 2024-11-23
> >
> > ***continue***
> >
> > - **Q4: The issue of representing the complexity of molecular data.**
> >
> >   This is an inspiring question. Linear models, despite their high explainability, potentially underfit complex molecular data. We address this by extracting informative LLM knowledge and using residual calibration to augment the linear model. Empirically, while others excel on simpler datasets like Mutag but falter on complex ones like Liver, *MoleX* achieves **13.2%** higher classification and **16.9%** higher explanation accuracy on Liver. This highlights *MoleX*'s capability of representing the complexity of molecular data.
> >
> > ---
> >
> > - **Q5: Comparison of explainability across *MoleX* and other interpretable models.**
> >
> >   Thanks for your question. As outlined in the experimental setup, we follow the milestone work GNNExplainer [1] and use AUC to compare the explainability of *MoleX* with other interpretable models on molecular data.
> >
> >   Besides, conventional ML models (e.g., linear regression) with mathematically derivable outputs are considered explainable [2], while neural nets are deemed unexplainable. We compare the performance of explainable and unexplainable models through extensive experiments, focusing on demonstrating *MoleX*'s superior explainability over other models without comparing the level of explainability among different interpretable models.
> >
> > ---
> >
> > ### References
> >
> > [1] Ying, Zhitao, et al. "GNNExplainer: Generating explanations for graph neural networks." *Advances in Neural Information Processing Systems* 32 (2019).
> >
> > [2] Hastie, Trevor. *The Elements of Statistical Learning: Data Mining, Inference, and Prediction.* (2009).

---

### Official Review · Reviewer_3Cvr · 2024-11-04

**Soundness:** 3
**Presentation:** 3
**Contribution:** 3
**Rating:** 8
**Confidence:** 2

**Summary:**

The article discusses a framework called MoleX, which aims to provide a granular interpretation of the impact of each feature on predictions. The framework uses a linear model to measure the contributions of decoupled n-gram features to molecular properties. It discusses the explainable component and residual component of a model and how they can be used to determine the contribution of each feature to the prediction.

**Strengths:**

This paper exhibits several notable strengths that contribute to its overall quality and impact.
- Well-Written and Clear Structure
  The author does an good job of discussing the main points in the article, providing a clear and concise overview of the framework, its theoretical background, and its advantages. The text is well-organized, making it easy to follow the author's reasoning and understand the key takeaways.
- Relevant Problem
  The problem addressed in the paper is relevant and timely, tackling a significant challenge in the field. The author provides a clear motivation for the research, highlighting the importance of understanding feature contributions in molecular properties.
- Advantages Over State-of-the-Art
  The proposed solution, MoleX, offers several advantages over existing state-of-the-art methods. By employing n-gram coefficients in a linear model, MoleX provides a more nuanced understanding of feature contributions, enabling researchers to better interpret the results. The author effectively highlights the benefits of MoleX, demonstrating its potential to improve the field.

**Weaknesses:**

- Limited Experimental Evaluation
  The paper's experimental evaluation is limited, with a narrow focus on a specific dataset and task. To further demonstrate the effectiveness of MoleX, it would be beneficial to conduct more extensive experiments on diverse datasets and tasks.
- Lack of Comparison to Other Methods
  The paper could benefit from a more comprehensive comparison to other state-of-the-art methods. While the author mentions some existing approaches, a more detailed analysis of their strengths and weaknesses would provide a clearer understanding of MoleX's advantages.
- Future Work
  The paper concludes somewhat abruptly, without discussing potential avenues for future research. Outlining possible extensions or applications of MoleX would provide a clearer direction for future studies and encourage further exploration of the framework.

**Questions:**

Can you discuss the potential limitations of the linearity assumption in MoleX? How might non-linear relationships between features impact the accuracy of the framework?

How do you interpret the n-gram features in the context of molecular properties? Can you provide examples of how specific n-grams might be related to particular molecular properties or mechanisms?

How robust is MoleX to different types and levels of noise? Are there any mechanisms in place to mitigate the effects of noise or outliers on the computed feature contributions?

You mention that "Theoretically, we provide a mathematical foundation to justify MoleX's explainability". Could you clarify what you mean by "theoretically" in this context? Are you implying that the mathematical foundation is theoretical in nature, or is this a statement about the theoretical soundness of the foundation?

---

> ### Author Response · Authors · 2024-11-23
>
> We sincerely appreciate your constructive feedback and your insightful concerns.  We are glad that you find our study **"well-written, well-organized"**, **"clear and concise overview of the framework, its theoretical background"**, **"a clear motivation"**, and **"better interpret the result"**. Below we will address your questions in detail.
>
> ---
>
> - **Q1: Limitation of linearity assumption.**
>
>   This is an insightful question closely tied to the core of our design. Indeed, the linearity assumption has two inherent limitations, which we address with proposed solutions:
>
>   - Linear models' lack of expressiveness can lead to underfitting on molecular data. We propose residual calibration to iteratively calibrate prediction errors made by the explainable model. Experiments show that the calibrator improves model performance by more than **12%**.
>   - *MoleX*'s success depends on the LLM's ability to capture significant features during linear model fitting. We thus propose information bottleneck-inspired fine-tuning and sparsity-inducing dimensionality reduction to ensure task-relevant information is maximally maintained for linear model fitting.
>
>   Our techniques resolve limitations and augment the linear model, with extensive experiments validating its effectiveness and robustness.
>
> ---
>
> - **Q2: Interpretation of n-grams.**
>
>   This is a good question. N-gram features represent **n** interconnected functional groups and their interactions, providing chemical contextual semantics about molecules. For example, in the Mutag dataset, 3-grams include functional groups such as benzene, nitro, and interactions such as pop and branch. This substructure has been validated to contribute to a molecule's mutagenicity.
>
> ---
>
> - **Q3: Robustness of *MoleX* to noises.**
>
>   We appreciate your inspiring question. Molecular data potentially contain noises [1-3], which we accounted for when developing *MoleX*. Additionally, noise can also arise from LLM embeddings. To mitigate this, we propose EFPCA and IB-inspired fine-tuning to eliminate noisy information, enabling the linear model to focus only on informative features and ensuring effective feature contribution computation. Experiments across ten datasets—spanning mutagenicity, carcinogenicity, drug toxicity, and permeability—consistently demonstrate *MoleX*'s robustness.
>
> ---
>
> - **Q4: Theoretical foundations.**
>
>   We thank your suggestion to clarify. The term "theoretically" refers to the theoretical analysis of *MoleX*'s explainability in the Appendix. This analysis is grounded in the mathematical foundations of linear models, convex optimization, and information bottleneck theories. These principles, while theoretical in nature, are adapted to our context through validated corollaries and lemmas, supported by mathematical proofs.
>
> ---
>
> ### References
>
> [1] Hillis, D. M., and J. P. Huelsenbeck. "Signal, noise, and reliability in molecular phylogenetic analyses." *Journal of Heredity* 83.3 (1992): 189-195.
>
> [2] Plesa, Tomislav, et al. "Noise control for molecular computing." *Journal of the Royal Society Interface* 15.144 (2018): 20180199.
>
> [3] Ioannidis, John PA. "Microarrays and molecular research: noise discovery?" *The Lancet* 365.9458 (2005): 454-455.

---

> ### Author Response · Authors · 2024-11-23
>
> ***continue***
> - **W1 and W2: More experimental evaluations on the effectiveness of *MoleX*.**
>
>   We appreciate your suggestions on our empirical results. In response to your feedback, we conducted additional evaluations of *MoleX* on three widely used datasets spanning diverse domains and tasks in molecular property prediction. The results of classification and explanation accuracy are shown as follows and have been incorporated into our updated manuscript.
>
> #### Classification accuracy over three datasets (%)
>
> | Methods                   | BBBP              | ClinTox           | HIV              |
> |---------------------------|-------------------|-------------------|------------------|
> | GCN                       | 78.5 ± 0.8       | 78.2 ± 1.0       | 72.1 ± 0.8       |
> | DGCNN                     | 80.0 ± 0.9       | 79.0 ± 1.1       | 73.2 ± 1.4       |
> | edGNN                     | 79.0 ± 0.9       | 77.5 ± 1.0       | 69.5 ± 0.7       |
> | GIN                       | 82.0 ± 0.7       | 80.9 ± 0.9       | 74.0 ± 1.3       |
> | RW-GNN                    | 81.0 ± 1.0       | 78.5 ± 1.0       | 75.5 ± 0.4       |
> | DropGNN                   | 83.0 ± 0.9       | 81.0 ± 0.8       | 64.5 ± 0.6       |
> | IEGN                      | 85.5 ± 1.0       | 80.1 ± 0.5       | 76.0 ± 0.9       |
> | LLAMA3.1-8b               | 69.0 ± 2.5       | 52.0 ± 2.7       | 56.0 ± 1.5       |
> | GPT-4o                    | 74.5 ± 2.3       | 56.4 ± 2.5       | 64.5 ± 1.8       |
> | ChemBERTa-2               | 78.0 ± 1.5       | 71.5 ± 1.4       | 73.0 ± 0.6       |
> | Logistic Regression       | 66.5 ± 0.8       | 60.2 ± 0.6       | 60.1 ± 0.7       |
> | Decision Tree             | 70.3 ± 0.8       | 62.8 ± 0.6       | 66.2 ± 0.8       |
> | Random Forest             | 73.5 ± 0.9       | 68.5 ± 0.7       | 69.8 ± 1.9       |
> | XGBoost                   | 74.2 ± 0.8       | 67.8 ± 0.8       | 70.2 ± 1.2       |
> | w/o Calibration           | 80.6 ± 1.3       | 85.9 ± 0.7       | 75.6 ± 1.3       |
> | **w/ Calibration (Ours)** | **93.1 ± 0.6**   | **94.1 ± 0.8**   | **81.3 ± 1.4**   |
>
> ---
>
> #### Explanation accuracy over three datasets (%)
>
> | Methods                   | BBBP              | ClinTox           | HIV              |
> |---------------------------|-------------------|-------------------|------------------|
> | GCN                       | 75.1 ± 0.4       | 74.6 ± 0.6       | 67.6 ± 0.6       |
> | DGCNN                     | 77.6 ± 1.1       | 79.2 ± 0.5       | 73.8 ± 1.1       |
> | edGNN                     | 78.9 ± 0.2       | 74.8 ± 0.2       | 71.6 ± 0.6       |
> | GIN                       | 80.4 ± 0.7       | 77.1 ± 0.8       | 70.3 ± 0.8       |
> | RW-GNN                    | 79.5 ± 0.4       | 69.4 ± 0.6       | 69.5 ± 0.7       |
> | DropGNN                   | 72.6 ± 0.6       | 76.7 ± 0.2       | 74.4 ± 0.3       |
> | IEGN                      | 80.8 ± 0.7       | 79.1 ± 0.4       | 69.5 ± 1.2       |
> | Logistic Regression       | 67.9 ± 0.3       | 61.9 ± 0.2       | 61.8 ± 0.6       |
> | Decision Tree             | 68.4 ± 1.5       | 66.8 ± 0.8       | 64.0 ± 1.2       |
> | Random Forest             | 73.3 ± 1.1       | 68.3 ± 1.7       | 65.7 ± 1.3       |
> | XGBoost                   | 73.5 ± 1.4       | 65.5 ± 1.6       | 67.8 ± 0.9       |
> | w/o Calibration           | 81.1 ± 1.8       | 78.6 ± 1.5       | 71.2 ± 1.1       |
> | **w/ Calibration (Ours)** | **90.8 ± 1.6**   | **92.8 ± 1.9**   | **82.4 ± 1.2**   |
>
> ---
>
> - **W3: Suggestion of future works.**
>
>   We apologize for the confusion. We have discussed potential future works at the end of the Appendix. This section includes the exploration of framework generalizability, the diversity of LLMs, and the trade-off between complexity and performance. We hope this discussion will shed light on further research based on our study.

---

### Author Response · Authors · 2024-11-23

We sincerely appreciate the reviewers' time and effort in providing valuable feedback. We have incorporated the suggested revisions into our manuscript, highlighted in blue.

We are glad that they find the problem we solve with **“a clear motivation/innovative”** (3Cvr, P2P1); our proposed method **“stands out for its innovative approach/improve the field/high interpretability/robustness/effectiveness”** (3Cvr, P2P1, wPWb, s7A7, CoqY) with **“creative solution/meaningful contribution”** (P2P1); our theoretical foundation **“clear and concise theoretical background”** (3Cvr), **“technical rigor, clarity, and significant practical implications”** (P2P1); our empirical analysis **“holistic experiment evaluation/exhaustive experiments/better interpret the result/granular interpretation”** (s7A7, CoqY, 3Cvr); as well as our presentation **“well-written/well-organized”** (3Cvr).

The specific changes made are detailed in our individual responses to the reviewers who requested them. The primary modifications in our submission include:

1. Elaborating on residual calibration with additional details (both high-level idea and empirical analysis).
2. Adding pseudo-code to illustrate the entire workflow.
3. Fixing and clarifying mathematical descriptors with further explanations.
4. Providing more empirical evaluations on robustness.
5. Detailing the experimental setup.

---

### Meta-Review · Area_Chair_vn82 · 2024-12-17

**Metareview:**

**Summary:** The authors use LLMs to extract the maximum amount of task-relevant knowledge from their dimensionality-reduced embeddings and fit a linear model for explainable inference. This is used to determine the contribution of each feature for prediction. They also introduce residual calibration to address prediction errors stemming from linear models' insufficient expressiveness of complex LLM embeddings. The faster performance of MoleX is demonstrated experimentally.

**Strengths:** The paper tackles an interesting problem that holds practical importance. It also provides an accurate yet highly interpretable approach. The main advantage seems to be using an LLM for feature extraction, which captures the higher-order interaction of the components. The method appears to outperform other techniques experimentally.

**Weaknesses:** The main shortcomings of the paper are its novelty and experimental evaluation. Specifically, the paper combines well-established techniques to present a new framework. However, it does not add much as a technical contribution. In terms of experiments, several reviewers point out the lack of comparison to previously known interpretable techniques. There are also some issues with the writing and notation.

**Decision:** Although the paper provides some interesting ideas, I believe the novelty is still the main shortcoming of the work. The authors should also try to justify the rationale for each design choice and perform separate ablations. Moreover, additional comparisons to previous techniques would also improve the work. Finally, the authors should verify that the notation and the presentation are clear and consistent. At this stage, I recommend rejection.

**Additional Comments On Reviewer Discussion:**

The authors clarify some of the misunderstandings by the reviewers and provide additional experimental results. The paper requires further efforts to improve clarity and notation. In terms of novelty, the paper still lacks sufficient technical contributions to be accepted.

---

### Decision · Program_Chairs · 2025-01-22

Reject